# Lidar measurements of thin laminations within Arctic clouds

Emily M. McCullough[1,*], James R. Drummond[1], and Thomas J. Duck[1]

[1]Department of Physics and Atmospheric Science, Dalhousie University, 6310 Coburg Rd., PO Box 15000, Halifax, NS, B3H 4R2, Canada

*Correspondence to:* Emily McCullough (emccull2@uwo.ca)

**Abstract.**

Very thin (< 10 m) laminations within Arctic clouds have been observed in all seasons using the Canadian Network for the Detection of Atmospheric Change (CANDAC) Rayleigh-Mie-Raman lidar (CRL) at the Polar Environment Atmospheric Research Laboratory (PEARL; located at Eureka, Nunavut in the Canadian High Arctic). CRL's time (1 min) and altitude
5   (7.5 m) resolution from 500 m to 12+ km altitude make these measurements possible. We have observed a variety of thicknesses for individual laminations, with some at least as thin as the detection limit of the lidar (7.5 m). The clouds which contain the laminated features are typically found below 4 km, can last longer than 24 h, and occur most frequently during periods of snow and rain, often during very stable temperature inversion conditions. Results are presented for range-scaled photocounts at 532 nm and at 355 nm, ratios of 532/355 nm photocounts, and 532 nm linear depolarization parameter, with context provided
10   by twice-daily Eureka radiosonde temperature and relative humidity profiles.

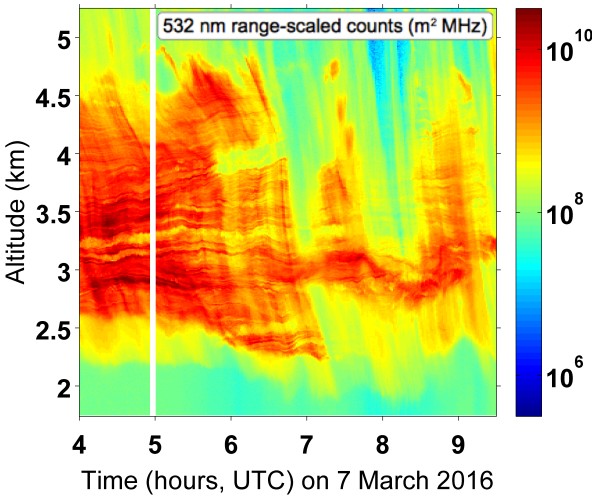

**Figure 1.** Thin laminated layers within an Arctic cloud. 532 nm range-scaled counts from the CRL lidar at Eureka, Nunavut showing quasi-horizontal layers, as thin as 7.5 m each, within a cloud on 7 March 2016, during snowing conditions.

## 1 Introduction

High resolution studies of clouds, and in particular Arctic clouds, are essential for a full understanding of the clouds' microphysical properties. Even if the clouds appear identical at low resolution, significantly different processes may occur in morphologically distinct clouds, e.g. a layered cloud in which the size of the layers is smaller than the resolution of the measuring instrument or model, and a smooth cloud with the same average optical properties as the layered cloud.

Figure 1 shows 532 nm range-scaled counts (counts $\times$ altitude$^2$) from the Canadian Network for the Detection of Atmospheric Change (CANDAC) Rayleigh-Mie-Raman lidar (CRL) at the Polar Environment Atmospheric Research Laboratory (PEARL; located at Eureka, Nunavut in the Canadian High Arctic). The figure shows quasi-horizontal layers, as thin as 7.5 m each, within a cloud on 7 March 2016, while snowing conditions were reported at the surface. CRL's highest resolution is required to resolve the thinnest laminations. There are descending features in Fig. 1 interpreted to be fall streaks. These do not seem to interfere with the persistence of the laminated features. There are at least 16 layers in the region between 3.25 and 3.75 km at 06:30 UTC, giving a mean layer thickness of 15 m. Some layers merge together into thicker layers, and split again into thinner layers, over the course of this 5.5 h plot. This example is not an isolated case. Similar phenomena are displayed frequently the CRL measurements, with individual cases often spanning several days in a row.

Figure 2 shows selected profiles of range-scaled 532 nm photocounts from Fig. 1 as a function of altitude for four consecutive minutes just after 06:40 UTC, each offset by $1\times10^{0.6}$ (or $4\,m^2$MHz) along the x-axis, between the altitudes of 3 to 4 km. There are clearly horizontal coherent structures in the cloud in space (aliased to time by motion over the lidar) at least down to the 7.5 m height resolution of the lidar. The regions between the laminations generally exhibit range-scaled signals between 35 and 70 % lower than the signals of the laminations immediately above and below.

If the data are averaged to altitude bins 10 times as large as those shown, all traces of the laminated structure would be erased (Fig. 3), and the cloud would look more similar to a smooth cloud. The higher resolution is required to have our interpretations approach a real representation of the cloud. Even in specific circumstances which could ensure that the layered cloud and the equivalent smooth cloud radiate equally overall, and thus influence the overall radiation budget in the same way, there is much to be learned about the disparate internal processes which form, maintain, evolve, and dissipate each of the clouds. Cloud-aerosol interactions, cloud condensation, particle growth, and precipitation are all microscale processes which may be better probed by measurements which can discern spatially inhomogenous cloud particle distributions from homogenous distributions. With a paucity of cloud measurements available in Arctic regions, as compared to mid-latitudes, high-resolution lidar measurements will be all the more valuable from polar laboratories.

High spatial and temporal resolution lidar measurements, particularly of cloud microphysical parameters, have been clearly stated in the literature as being desirable and necessary. The vertical size scales deemed to correspond to "high enough" spatial resolution, vary. Mioche et al. (2017), Loewe et al. (2017), and Hogan et al. (2003), make the case for sub-100 m sampling. Ramaswamy and Detwiler (1986), Korolev et al. (2007), Sotiropoulou et al. (2014) and Solomon et al. (2015) are several examples advocating for measurements at sub-50 m resolution. The current paper is concerned with measurements at sub-10 m scales.

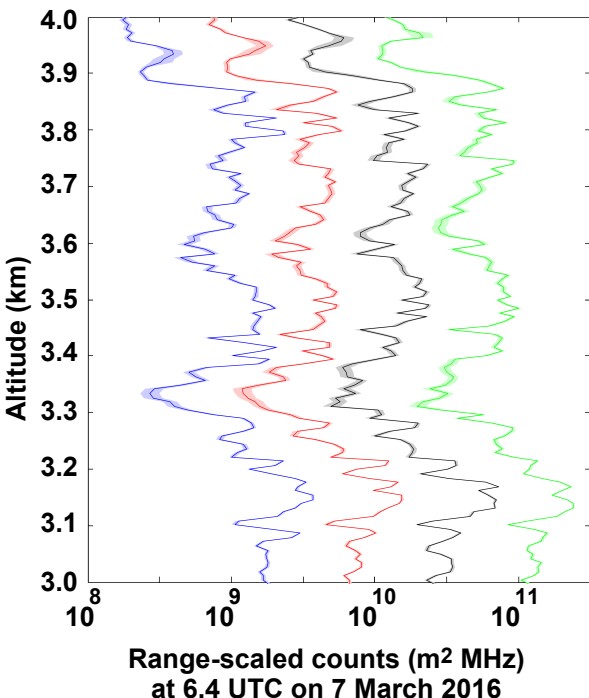

**Figure 2.** Selected profiles of range-scaled 532 nm photocounts as a function of altitude for four consecutive minutes just after 06:40 UTC on 7 March 2016 (same date as for Fig. 1), each offset by $1x10^{0.6}$ (or $4\,m^2\,MHz$) along the x-axis, between the altitudes of 3 to 4 km. Shaded areas show uncertainty. There are clearly horizontal coherent structures in the cloud in space (aliased to time by motion over the lidar) at least down to the 7.5 m height resolution of the lidar.

The literature, also, has many reports of vertically "narrow" or "very thin" measured features. These come at a large range of spatial sizes, generally larger than the scales that we are interested in here. Mid-latitude examples of "notably thin" features include: Sassen et al. (2005), who describe a "remarkably narrow" feature (a dark-(lidar) and bright-(radar) band attributed to regions of snowflake melting) with a full width half maximum (FWHM), estimated from their Fig. 4, of approximately
5   500 m. Since a resolution of 75 m was used, higher resolution features should have been detectable had there been any present. Hayman et al. (2012) used a higher resolution lidar (7.5 m x 0.5 s) in Boulder, Colorado, USA to detect a "narrow altitude band" of differently oriented scatterers which extends between 5 and 5.5 km, and therefore is 500 m in vertical extent. Hogan et al. (2003) ran aircraft measurements over the UK, with some analysis possible at 15 m resolution, and they describe "thin layers of high [attenuated backscatter coefficient] around 150 m thick", and others 100 m to 200 m thick.
10   We have been unable to find many references to cloud features at sub-100 m scales in the literature. Indeed, it is difficult to find any reference to multiple layers within clouds (as in Fig. 1) as opposed to multiple layers of clouds (2 or 3 separate clouds at different altitudes, separated by hundreds of metres to several kilometres, e.g. Curry et al. (1988)). Likewise, thin (100 - 200 m thick) layers of supercooled liquid water are known to frequently top mixed-phase clouds, generally precipitating

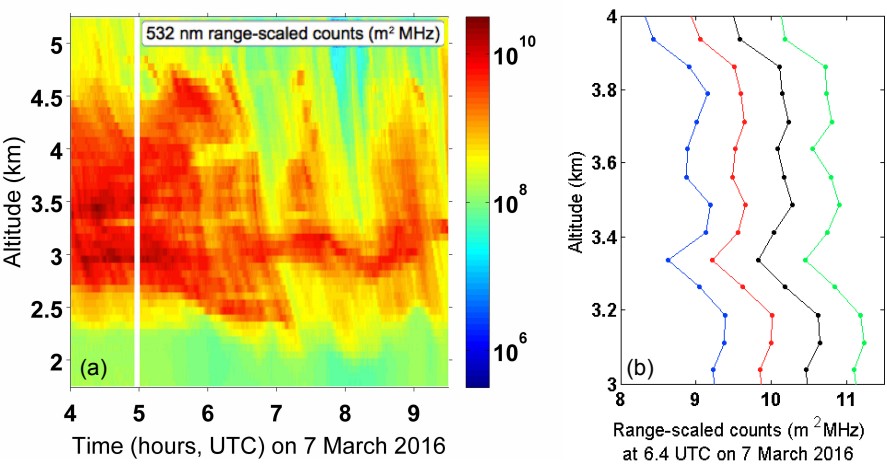

**Figure 3.** Same measurements as Figs. 1 (a) and 2 (b), recalculated with altitude bins 10 times as large. Resolution is 1 min X 75 m. The fine (< 10 m) laminations are no longer discernible. Only the much larger features (e.g. between 3.2 and 3.4 km) remain.

ice (Morrison et al., 2012; Shupe et al., 2008). Again, these situations are quite different in morphology from the laminated features described in this paper.

Measurements by airborne holographic imaging have visualized the spatial structure in clouds at centimetre scales by measuring droplet size and number distributions, revealing that clouds are inhomogeneous and contain sharp transitions between

cloud and clear air properties even at the smallest turbulent scales (Beals et al. (2015)). Given that there are "edges" within clouds even at cm scales, it is reasonable to infer that there may be structural cloud features which are relevant to the overall interpretation of particular clouds, which are possible to investigate by lidar at resolutions of tens of metres and which will be missed entirely by lidar measurements at 100+ m scales. Certainly, the scales probed in Beals et al. (2015) are significantly smaller than those possible to investigate using the CRL lidar. Cloud measurements covering the entire range of spatial scales

from centimetre to global is ultimately required. CRL helps close the gap from over four orders of magnitude of spatial size, to three, between the holographic imaging measurements and the smallest features currently discussed in the lidar literature.

The closest description that we have found to the laminations, and which indeed may show the identical phenomenon, comes from Hobbs and Rangno (2008), with cloud particle concentration and size measurements from airborne campaigns over the Beaufort Sea in April 1992 and June 1995. Their vertical profiles of cloud droplet concentrations show "adjacent

layers, separated by only tens of metres ... often exhibiting substantially different droplet concentrations". They infer that the layers are not mixing with one another, and note that more non-mixed clouds are observed than mixed ones during the campaigns. Their Fig. 4 is demonstrates these layers. Like CRL's results, the horizontal flight path of the aircraft aliases spatial and temporal phenomena somewhat: "In some cases cloud layers separated by short distances merged together for a time", as indicated by the aircraft flying into a sudden region of increased liquid water content. CRL sees something similar, with

individual layers seeming to merge and separate along the time axis of the photocount plots. Hobbs and Rangno (2008) note

multiple temperature lapse rates within single clouds, usually including regions of stability. Slight stability is noted as a cause for non-mixing in some cases, but is not present in all non-mixed (multiple-layered) cases. This leaves open some room for investigation into the mechanisms of formation and persistence of the layers.

If we extend our search to include studies of Arctic haze, more numerous results are available at high vertical resolution, and references are made to thin layers within a particular single unit of haze. There was a Mie lidar present at Alert, Nunavut, Canada for 9 weeks in 1984-5 (Hoff, 1988) for the purpose of studying the vertical distribution of Arctic haze. Its 694.3 nm laser with 4.6 m maximum vertical resolution measured layers as thin as 100 m in several cases, but none of these had the laminated morphology seen by CRL. Several aircraft campaigns have shown stacked haze layers on the order of tens of metres thick. Radke et al. (1989) used a 1064 nm downward-pointing aircraft lidar with resolution 3 m vertically x 40 m horizontally. It flew for two days in March 1986, ending in a polar airmass over Baffin Island which contained thin layers of haze. They are described as "multiple thin, discrete laminae. Some of the hazes observed by us in the Arctic have been < 20 m thick". These features approach the same order of magnitude as the cloud features observed by CRL which are presented in the present paper. Brock et al. (1990) made a flight one month later in April 1986 between Thule, Greenland, and Søndre Strømfjord, Greenland. The results include multiple thin haze layers of thickness between 30 and 60 m, separated by regions of similar thickness of cleaner air. These campaign results were confirmed a decade later by Khattatov et al. (1997), who ran an extended aircraft campaign and again found highly stratified haze over not only the Canadian Arctic, but over Russia and Germany as well. Figure 2 of Morley et al. (1990), which measured using 3 m and 7 m resolution modes, provides a plot which is strikingly similar to many shown later in the current paper. The differences are that while Morley et al. (1990) shows laminated aerosol layers 200 to 300 m thick, the CRL measurements are of laminated cloud layers which are closer to 10 m thick, and which are thus an order of magnitude smaller. All of the laminated haze layer reports are from aircraft campaigns of short duration, and all excluded from consideration any measurements which included ice crystals and clouds.

In mid-latitude examples of extremely strong atmospheric boundary layer stability, striations of fog may be identified at scales smaller than 1-metre (Mahrt (2014), Fig. 3). These are qualitatively similar to the cloud laminations identified by CRL. Perhaps the two phenomena share similar properties, particularly in terms of the factors which enable the laminations/striations to persist.

There is room for further investigation of clouds by lidar at size scales of tens of metres and smaller. The measurements presented in this paper begin to fill this gap in our measurement record, and demonstrate that finely laminated cloud features are present in Arctic clouds in the Canadian Arctic at all times of year. The laminated haze layers described in the literature are qualitatively similar in appearance to, and thus may share similar origins or mechanisms of persistence with, the laminated cloud layers presented here from CRL.

## 2  The CRL Lidar at Eureka, Nunavut

The Canadian Network for the Detection of Atmospheric Change (CANDAC) Rayleigh-Mie-Raman lidar (CRL) makes observations at the Polar Environment Atmospheric Research Laboratory (PEARL) at Eureka, Nunavut in the Canadian High Arctic ($80°$ N, $86°$ W).

CRL makes measurements at high resolution in altitude (7.5 m) and time (1 min) from 3.75 m to 120 km altitude. Above about 60 km, the lidar receives photons only from the sky background (scattered sunlight, moonlight, etc). Most of the signal from laser photons which are scattered by cloud and aerosol particles return from altitudes less than about 30 km. With analyses carried out CRL's highest resolution, retrievals are available from 500 m to 12+ km altitude. With overlap corrections, retrievals below 500 m are possible (Rotermund et al., 2014). Using coadding of signals (i.e. lower spatial or temporal resolution),

retrievals to higher altitudes (e.g. 20+ km) are routinely available (e.g. Zhao et al. (2014) and Lindenmaier et al. (2012)). See Nott et al. (2012) for a description of CRL and McCullough et al. (2017) for an updated description of its depolarization system. The relevant measurement channels for the present paper are the 355 nm Rayleigh elastic channel, the 532 nm Raleigh elastic channel, and the 532 nm depolarization channel.

## 3  Data reduction

Low-level data corrections as in McCullough (2015) and McCullough et al. (2017) have been applied to all raw photocount measurements. Namely, all photon counting data have been dead-time corrected and background subtracted; all analogue data have been dark count profile corrected, have been mapped from unitless measured values to the corresponding photomultiplier (PMT) voltages based on hardware settings, have been background subtracted, and have been converted from units of mV to equivalent photon count rates using gluing coefficients found during calibrations; the photon counting and analogue signals

have been merged together to create a single profile of photon count rate over all available signal levels for each channel. This value is expressed in MHz, which indicates the measured signal rate for each altitude bin, for each profile.

Typically, CRL data would be binned by co-adding in either altitude or time. This increases the signal to noise ratio (SNR) of the measurement, at the cost of reducing its resolution. For all plots in this paper, no post-integration of lidar photon counts was performed. We keep maximum resolution, at the cost of having some somewhat noisier plots at the higher altitudes. This

enables us to locate features with sizes on the order of one altitude bin (provided they last some time), or one time bin (provided there is some extent in altitude) for further study.

The 532 nm and 355 nm measured signal rates are multiplied by the square of the altitude of each data point to remove geometric altitude bias from the plots. The resulting range-scaled photocounts are then plotted on a logarithmic scale. Examples of such plots are given in Fig. 1, and in panels a, b, and c of Figs. 4 and 6. The range-scaled photocount plots have not been

normalized for laser power fluctuations, which are expected to remain $\leq 5\%$. Therefore, we can trust relative signal variations within each vertical profile of a plot more strongly than we can trust relative signal variations in time. One notable exception is the region below about 750 m altitude which is the region of incomplete geometric overlap for CRL. No overlap corrections have been made, so signals below this altitude may not be properly normalized with respect to the rest of the profile.

The second type of plot presented in this paper is a ratio of 532 nm to 355 nm measured signal rates. This is not the traditional 'colour ratio' sometimes published in lidar literature, since it is directly the ratio of signal rates, and is not a ratio of calibrated backscatter coefficient values. Examples of these plots are Figs. 4d and 6d.

The third type of plot in this paper is 532 nm linear depolarization parameter, calculated as per the $d_1$ method from Mc-Cullough et al. (2018): $d = (2kS_\perp)/(S_\parallel + kS_\perp)$. $S_\perp$ is the signal measured by the perpendicular channel, $S_\parallel$ is the signal measured by the parallel channel, and $k$ is the depolarization calibration constant ($k = 21$ for CRL). The depolarization may also be expressed as the depolarization ratio, which can be calculated directly from depolarization parameter: $\delta = d/(2-d)$. At CRL, the parallel and perpendicular channels share a single PMT. A Polarotor rotating prism with timing electronics admits received photons to each measurement profile on alternate laser shots. Examples of depolarization parameter plots are Figs. 4e and 6e. Appendix A provides some plots of depolarization uncertainty in Figs. 9 and 10.

Temperature and humidity profiles obtained using radiosondes launched from the Eureka Weather Station are also provided. No additional corrections have been made before plotting. The relative humidity values are plotted with respect to both water and ice in all cases. Examples of these plots are Figs. 5c,d and 7c,d.

## 4 Results

CRL made 182 days of measurements between March and December 2017. Of these, at least 45 days show evidence of horizontal laminations within clouds. Thus, laminations occurred on 25 % of all measurement dates. A minimum of one detection of laminations was present in each measured month. Hence, this phenomenon is not restricted to a particular season. March 2017 had highest rate of detections, with at least 10 of 24 measurement days demonstrating laminations. Three representative examples will be shown in full here: 21 March 2017 is in Section 4.1, 14 November 2017 is in Section 4.2, and 26 August 2017 is in Appendix C.

### 4.1 Layers present for 24 hours on 21 March 2017

On 21 March 2017, the 532 nm range-scaled counts show thin layers persisting through a 2 km thick cloud which is present for about 21 h as shown in Fig. 4a. The clouds began on the previous day (08:15 UTC 20 March 2017), and continued for another 2 h on the following day (until 02:00 UTC 22 March 2017). The portion of Fig. 4a inside Box A has been reproduced in a larger format for Figs. 4b,c,d,e, to show detail. Resolution for all colour plots is 1 min x 7.5 m. No further binning was performed.

Figure 4b is 532 nm range-scaled counts, and we can discern layers of several thicknesses within this area. The layers are quasi-horizontal, but can move vertically by small amounts (usually less than 50 m) over hours-long timescales. Below 1 km at 22:00 UTC there are some layers approximately 45 m thick. At 1.25 km at 23:00 UTC, there are layers 22.5 m thick interspersed with the thicker layers. A grouping of 4 layers is particularly noticeable at 2 km at 22:30 - 24:00 UTC, each layer having a thickness of 15 to 22.5 m. Many other thin layers are also present within this plot. Similar plots were examined at 2 x 2 data binning (to a resolution of 2 min x 15 m; not shown). As expected, all layers thicker than 7.5 m were still visible, but their edges were less well-defined. The 7.5 m thick layers were sometimes still visible, and sometimes not, with longer-lasting

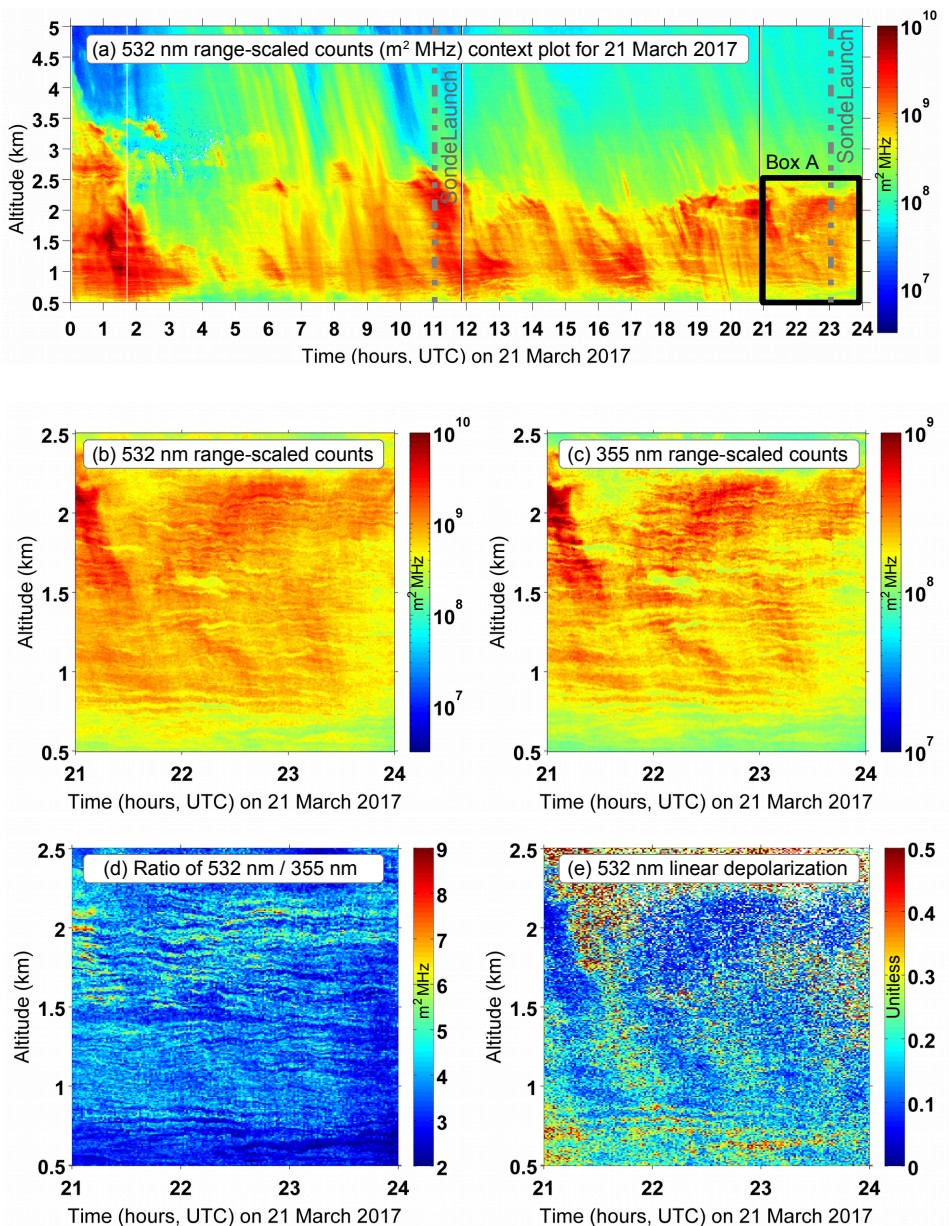

**Figure 4.** Measurements from 21 March 2017. Clouds persisted for the majority of the day, with thin layers visible in all clouds below 3 km altitude. Fall streaks indicative of precipitating particles are frequently present. This instance of laminated cloud lasted in excess of 42 hours, beginning on the previous day, and ending on the following day. **(a)** is a context plot of 532 nm range scaled photocounts. **(b, c, d, e)** are detailed plots for the region indicated by the black Box A of **(a)**. **(b, c)** are 532 nm and 355 nm range scaled photocounts, respectively; **(e)** is the ratio of 532/355 nm photocounts; **(e)** is the 532 nm linear depolarization parameter.

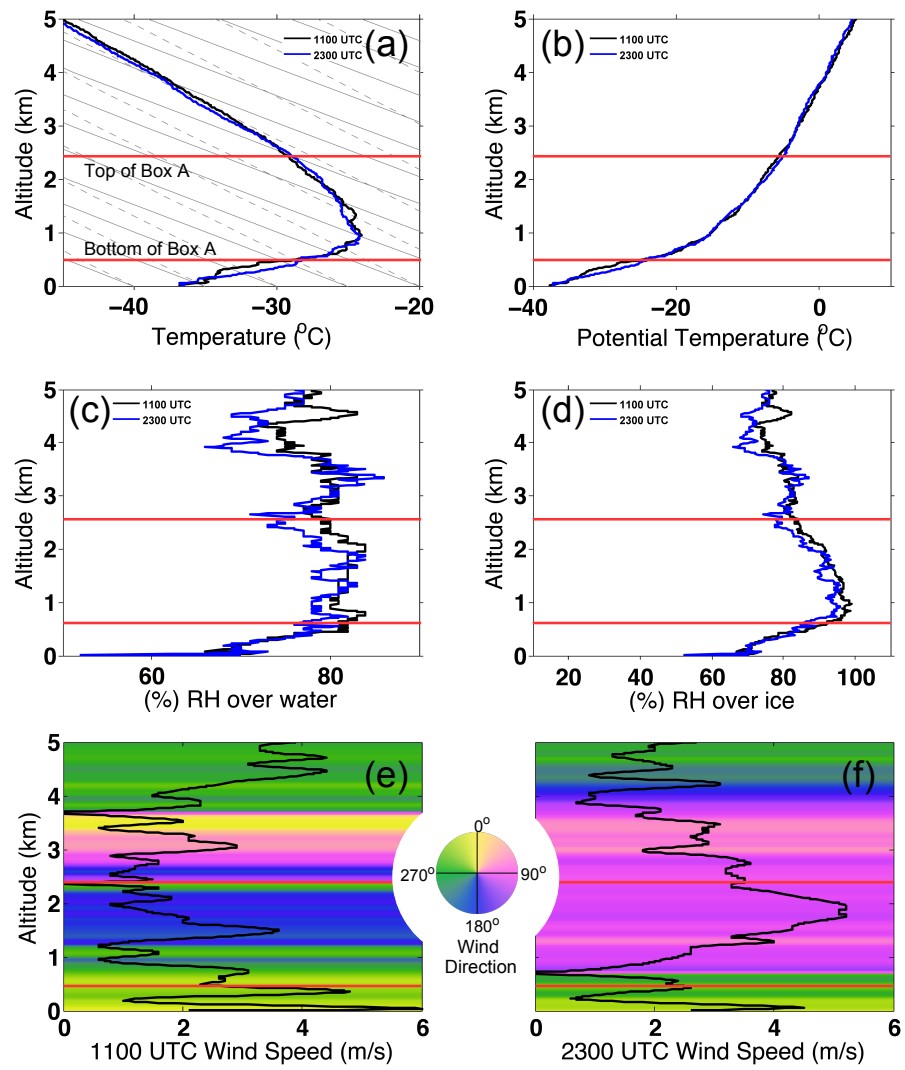

**Figure 5.** 21 March 2017. From the two daily rasiosondes launched by the Eureka Weather Station: **(a)** is the temperature; grey solid lines are dry adiabats, and grey dashed lines are moist adiabats. **(b)** is the potential temperature. **(c,d)** are the relative humidity with respect to water and ice respectively. **(e,f)** are the wind speed (black line) and direction (coloured background). Red lines show top and bottom altitudes of Box A from Fig. 4.

layers being easier to see. Several instances of fall streaks are visible within the plot, apparent from their descent in time. The fall streaks do not seem to prevent the continuation of the laminations within the cloud.

Figure 4c is 355 nm range-scaled counts. All bright layers visible in the 532 nm plot are also visible as enhancements in the 355 nm plot. The 355 nm channel has lower overall photon count rates than the 532 nm channel, so some of the weaker layers in terms of backscattered photon amplitude are not picked up in the 355 nm plot. For example, there is a pair of 7.5 m thick layers at 1.8 km just after 02:00 UTC which are seen in the 532 nm plot, but not in the 355 nm plot. All layers to which the 355 nm plot is sensitive are present also in the 532 nm plot.

Fig. 4d is the ratio of the 532 nm/355 nm MHz count rates. We see many of the same thin layered features in this type of plot. The thicker 45 m layers are clearly seen, as well as most of the brighter layers above 1.5 km which are thicker than 22.5 m. Layers as thin as 7.5 m which were identified in the individual plots for 532 nm and 355 nm can be found in this ratio plot, but they are not so obvious. This is not a traditional colour ratio, since it is taken between the count values themselves, and not between backscatter coefficient values. Nevertheless, the presence or absence of layers in the ratio plot, which are present in the individual plots, can provide extra information about the geophysical phenomena which form the layers. For certain particle size distributions, we may expect not to see the layers in such a calculation, despite their presence in the atmosphere. A more sophisticated approach to a colour ratio has been used to combine CRL measurements with radar measurements in Bourdages et al. (2009), but the resolution of the available radars at Eureka is not sufficient to resolve the 7.5 m features we see here.

Figure 4e is the 532 nm linear depolarization parameter. This is calculated using the $d_1$ method from McCullough et al. (2018), which is the technically simplest method to calculate the desired quantity. The downside of the method is that one of the measurement channels has very low signal rates, leading to a generally low signal to noise ratio (SNR). Consequently, the depolarization plot shown here is noisy, and the layers are difficult to discern. The 45 m thick layers are displayed with a high depolarization parameter, which indicates non-spherical particles. Typically, this means randomly oriented frozen particles within clouds, or aerosol particles outside of clouds. There are some small features which have higher depolarziation parameters than the surrounding areas, but which do not correlate with the layers seen in 4a,b,c. For example, the $d_1 = 0.25$ feature just below 1.5 km altitude which rises slightly between 21:00 and 21:30 UTC, and the parallel line about 0.2 km below it. The regions between the layers of high 532 nm backscatter, therefore, are the regions consistent with an interpretation of ice or aerosol particles. The regions within the high backscatter layers are not. The largest blue swathes in the depolarization plot correspond to general regions of the highest photocount rates in the 532 nm plot. The depolarization values in these regions are low and therefore combined with the high backscatter signal are consistent with liquid water droplets and/or preferentially oriented ice particles, and are inconsistent with interpretation as randomly oriented ice particles.

Although the depolarization plots are somewhat noisy at this resolution, absolute uncertainties are generally between 0.05 and 0.1 (in the same units as depolarization parameter) for the region below 1 km, where the laminations are visible in Fig. 4e. At higher altitudes, uncertainties for this date reach 0.16.

Figures 5a and b display measurements of temperature and potential temperature, respectively, from Eureka Weather Station radiosonde flights which took place at 11:00 and 23:00 UTC. The sonde data is plotted from 0 to 5 km to provide context for the plots in Fig. 4. The red lines on all Fig. 5 plots indicate the upper and lower altitude bounds of Box A from Fig. 4a, which are

also the altitude bounds of Figs. 4b,c,d,e. The 23:00 UTC flight falls within the time range of Figs. 4 b,c,d,e. Dry and saturated adiabats, in solid and dashed grey, respectively, provide a guide to the thermal stability within the cloud.

Figure 5a shows a strong temperature inversion whose temperature starts at -36° C at the ground, increasing to -28° C by the bottom edge of Figs. 4b,c,d,e, to a maximum temperature of -24° C at 1 km altitude, before the temperature starts decreasing throughout the troposphere. By the top edge of Figs. 4b,c,d,e the temperature has decreased to -29° C, and by the top edge of Figs. 4a at 5 km altitude, the temperature is -46° C. Some background information regarding temperature inversions for the Arctic is available in Lesins et al. (2012). Even above the temperature inversion thermal maximum, the air remains very stable, as indicated by comparison with the adiabatic lapse rates. The temperature profiles for both sondes are quite similar in shape. Figure 5b shows potential temperature for both sondes smoothly increasing from the ground to 1 km at a rate of about 22.2° C/km, and at a larger rate of 5.0° C/km at higher altitudes.

Figures 5c and d give the relative humidity over water and over ice (See Appendix B), respectively, for both sondes. Through the regions of 4b,c,d,e, relative humidity over water varies between 75 % and 85 %, while relative humidity over ice varies between 85 % and 97 %. Through the full region plotted in Fig. 4a, relative humidity over water remains between 65 % and 85 %, and relative humidity over ice remains between 85 % and 97 %. In both cases, the relative humidity increases very quickly from the ground up to 750 m altitude, before levelling off for relative humidity over water, and ultimately decreasing for relative humidity over ice. The relative humidity plots are relatively constant from sonde to sonde on this day.

Hourly metorological observations recorded by the Eureka Weather Station on 21 March 2017 note precipitation at ground level throughout the day: ice crystals at 00:00 UTC and 01:00 UTC, snow at 02:00 UTC through 12:00 UTC, and ice crystals again thereafter. The temperature recorded at the weather station varied between -35.7° C and -37.9° C during this time.

To explore the dynamics, Figure 5 shows wind speed (line plot) and direction (coloured background) in panels e and f from the 11:00 UTC and 23:00 UTC sondes. The wind profiles differ considerably in direction between the two sondes, although the magnitude of the windspeeds are of the same order of magnitude: between 0 and 5 m/s. The wind direction is much more variable in height for the 11:00 UTC sonde. Below 1.25 km, the wind is generally around 280°, at which point it rotates to about 180° until 2.5 km, then to 90° until 3.5 km, before returning by 4 km to a direction of 270°. Throughout the profile, there are small altitude ranges (e.g. at 2.3 km) which show larger windshear, but generally the change in wind direction is relatively gradual, with few complete reversals of direction. The 23:00 UTC profile, conversely, begins with wind direction at the same 280° direction from the ground to 750 m, then reverses to about 100° and remains constantly from this direction until 4 km, at which point it returns again to 260°. There are no instances of quick oscillations of wind direction with altitude in the 23:00 UTC profile which would be similar to those in the 11:00 UTC profile. In the region of Figs. 4b,c,d,e, the 23:00 UTC wind profiles are relevant from 0.5 to 2.5 km, a region which includes one reversal in wind direction just below 1 km altitude. At this location, the wind speed reaches zero as it changes from a generally decreasing profile in one direction to a generally increasing profile in the opposite direction. Around 2 km, the windspeed reaches a maximum of 5 m/s and then decreases.

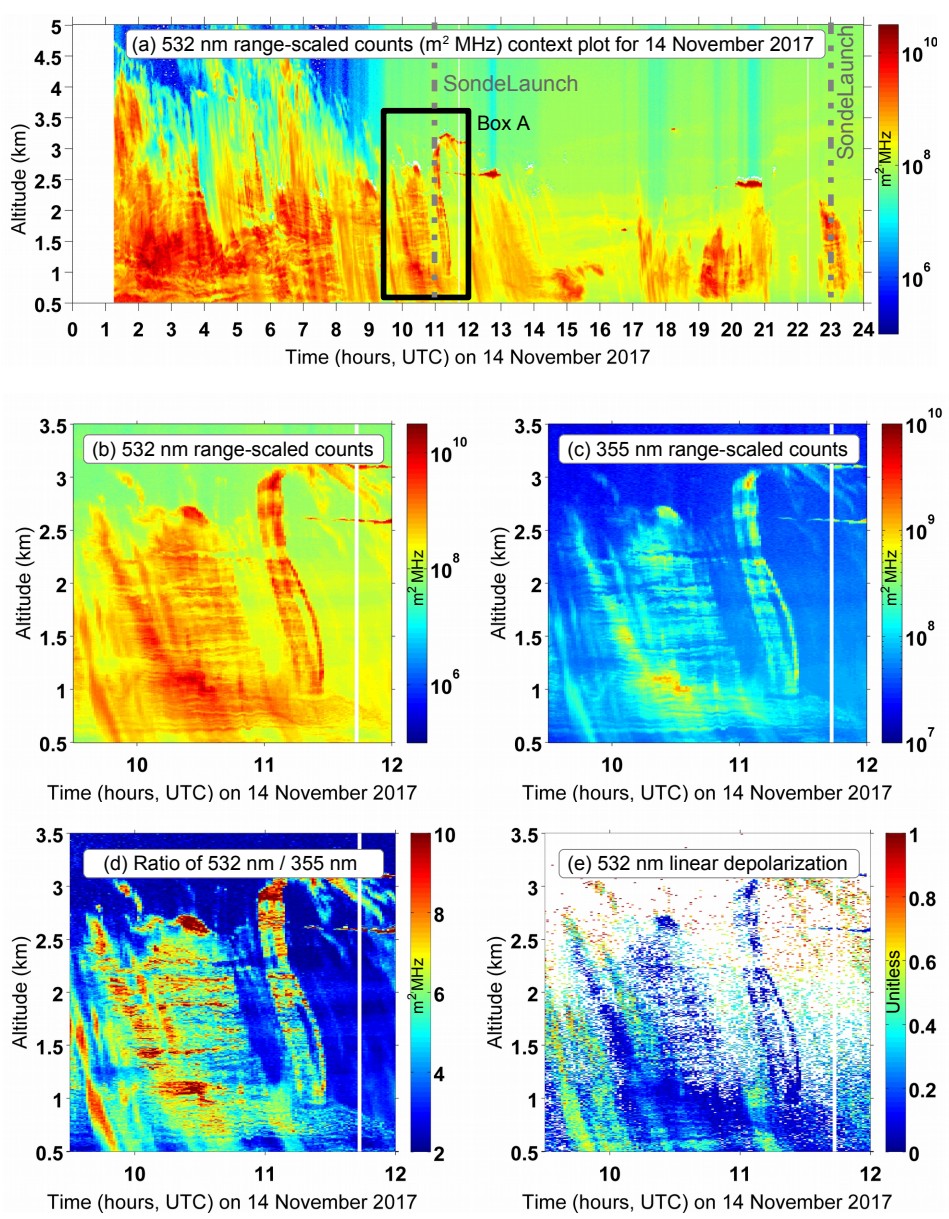

**Figure 6.** Measurements from 14 November 2017; **(a-e)** same format as Fig. 4. Thick clouds were present early in the day, with cloud cover reducing later. Layers which start in a cloud continue in the next section of cloud, even if there is a gap in between. Precipitation alternated between light snow, blowing snow, ice crystals, and no precipitation at the ground throughout the day.

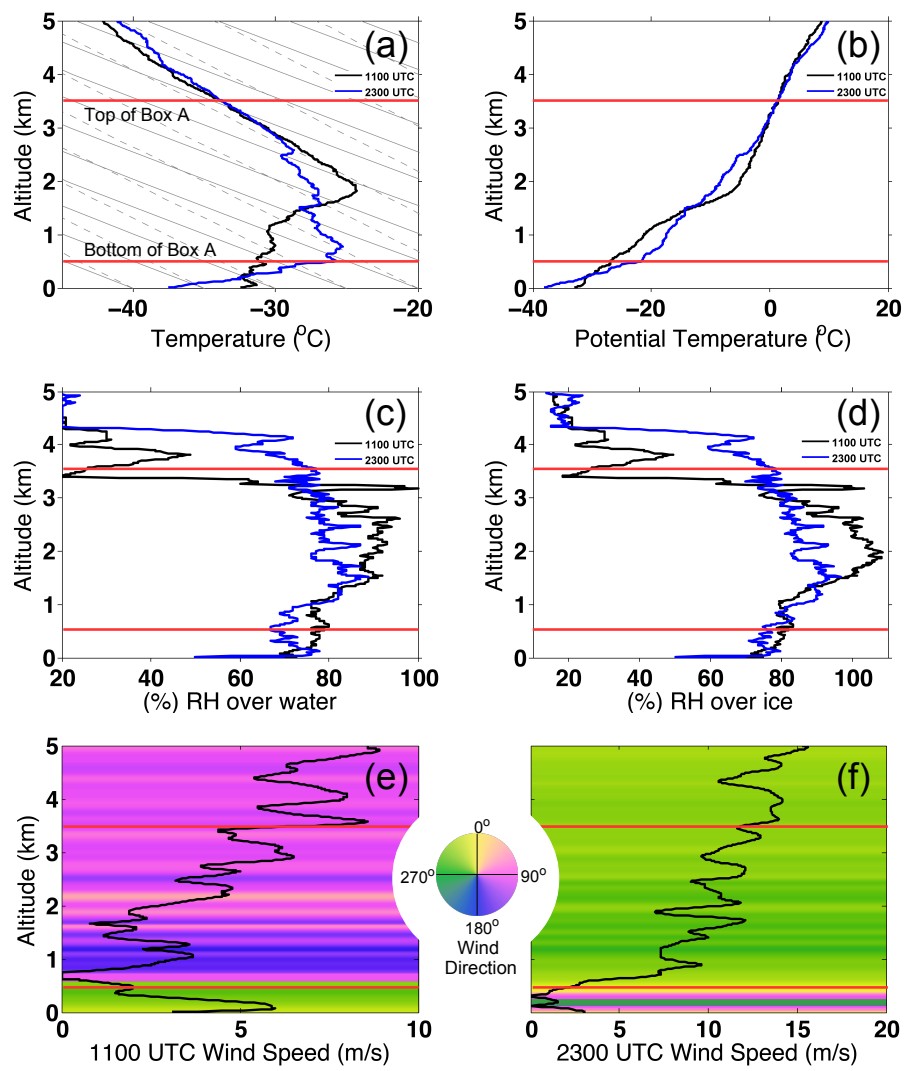

**Figure 7.** 14 November 2017; **(a-f)** same format as Fig. 5. Red lines show top and bottom altitudes of Box A from Fig. 6.

## 4.2 Layers reappearing several times on 14 November 2017

On 14 November 2017, the 532 nm range-scaled counts in Fig. 6 show thin layers similar to those in the 21 March 2017 example (Fig. 4). The clouds which contain the layers are slightly different. The day begins with clouds thicker in vertical extent (4.5 km rather than 3.5 km), with peak count rates 3 times larger ($1 \times 10^{10.5}$ m$^2$MHz rather than $1 \times 10^{10}$ m$^2$MHz; equivalent to $3.2 \times 10^{10}$ m$^2$MHz vs. $1 \times 10^{10}$ m$^2$MHz). There is some internal layering during the cloud from 01:00 to 08:00 UTC with layers on the order of 7.5 m up to 50 m thick. This thick cloud lasts until about 08:00 UTC, at which point it diminishes drastically in optical thickness, and then becomes discontinuous for the rest of the day. The thinner, patchy clouds after 12:00 UTC are restricted to altitudes below 2.5 km.

Layers which start in a cloud continue in the next section of cloud, even if there is some non-cloudy region in between. The layers seem contiguous. The layers seem to continue between periods of fall streaks indicative of precipitating particles. Around 11:00 UTC at 1.3 km, 1.6 km, and two layers near 2 km, we can see some remnants of these layers with photocount values that would seem to indicate aerosols, and not cloud particles, between the obvious clouds. This is more apparent in the 532 nm and 355 nm range scaled counts plots when the colourbar is rescaled (not shown), and can be seen in the colour scale for the ratio 532/355 nm plot in Fig. 6c.

The plots of 14 November 2017 are a good example of a day which has both layered clouds (01:00 - 11:30 UTC; 22:30 - 23:30 UTC) and clouds without layers (12:00 - 21:00 UTC).

Some of the layers are visible in the depolarization parameter plot, Fig. 6d, but not all of them. This is likely to be a sensitivity issue in some regions, as we are operating at the detection limit of the depolarization's perpendicular measurement channel. In other regions, such as in the prominent fall streak visible as bright green at the bottom left corner of the plot, extending from 01:30 km at 09:30 UTC to 0.5 km before 10:00 UTC, sensitivity is unlikely to be the reason that the layers are not visible. There, since backscatter is high, and depolarization $d = 0.5$ is high also, precipitating frozen particles are a reasonable interpretation. We do not see any layering in this type of feature in any of the plots. For the regions in which we do see laminated depolarization, the depolarization parameter is anticorellated with photon count rate at both wavelengths in Fig. 6. The depolarization parameter is low (values of less than 0.1, dark blue in Fig. 6e) when the count rates in both the 532 and 355 nm channels are high ($1 \times 10^{8.8}$ (or $6.2 \times 10^8$), red in Fig. 6b, and $1 \times 10^{8.5}$ (or $3.2 \times 10^8$), yellow in Fig. 6c, respectively). One particular layer which demonstrates this quite clearly is at 0.6 km altitude, from 10:30 UTC - 10:45 UTC. This layer is dark blue (low values) in the depolarization parameter plot, but red and yellow (high values) in the 532 nm and 355 nm range scaled counts plots. Corresponding 532/355 nm values are also high. Therefore, as for the 21 March 2017 example, we interpret the laminations with high backscatter and low depolarization to be most likely liquid particles, and unlikely to be aerosol or ice. Conversely, the spaces between the high backscatter laminations exhibits higher depolarization which, combined with low backscatter values, leads to a reasonable interpretation of aerosol particles.

For Fig. 6e, the uncertainties are somewhat higher than they are for 21 March 2017 (4e) in regions of high depolarization, reaching values of 0.2 to 0.3 where $d > 0.5$. Similar to the March example, regions on 14 November 2017 in which cloud

laminations are visible, namely between 10:30 and 11:00 UTC below 1 km, have absolute uncertainties smaller than 0.06 in general.

Figures 7a and b display measurements of temperature and potential temperature, respectively, from Eureka Weather Station radiosonde flights which took place at 11:00 and 23:00 UTC. The sonde data is plotted from 0 to 5 km to provide context for the
plots in Fig. 6. The red lines on all Fig. 7 plots indicate the upper and lower altitude bounds of Box A from Fig. 6a, which are also the altitude bounds of Figs. 6b,c,d,e. The 23:00 UTC flight falls within the time range of Figs. 6 b,c,d,e. Dry and saturated adiabats, in solid and dashed grey, respectively, provide a guide to the thermal stability within the cloud.

Radiosonde temperatures in Fig. 7a at 11:00 UTC show a temperature inversion which begins at -32° C at the ground, increasing slowly in temperature to -30° C at 1.25 km, increasing then quite steeply to -24° C at 1.75 km (which is about the
middle altitude of Figs. 6b,c,d,e). The temperature then decreases linearly to -34° C at 3.5 km (top of the small plots), and continuing the linear decrease to -43° C by 5 km. The temperature fluctuations shown by the sonde are large in the lowest altitudes, on the order of 1° C. The 23:00 UTC curve is quite different from the 11:00 UTC curve for 14 November 2017, showing an even stronger temperature inversion from -37° C at the ground to -25° C at 900 m, followed by a slow decrease to -29° C at 3 km. The temperature curve matches that from the 11:00 UTC sonde between 3 and 5 km.

The potential temperature profiles in 7b are more similar for the two sondes on this date. Both following a general increase from the ground to 5 km altitude. The slopes are slightly different for each sonde: For 11:00 UTC, potential temperature increased at a rate of 30.5° C/km for the first 550 m, then increased at a lower rate of 6.9° C/km until 5 km. The 23:00 UTC sonde found potential temperature to rise at a rate of 10.75° C/km for the first 1.2 km, which then increased to a rate of 17.9° C/km until 2 km, and then decreased to 4.8° C/km until 5 km. Between 2.75 km and 5 km the potential temperatures for
both sondes are nearly identical.

Figures 7c and d give the relative humidity over water and over ice, respectively. The overall shape of the curves are quite similar for each, with the major difference being that the profiles for relative humidity over water do not exceed 100 % at any point, while the 11:00 UTC plot for relative humidity over ice does, within the altitude range of the cloud. Radiosonde relative humidity with respect to water in Fig. 7c is relatively constant with altitude for both sondes up to 1.25 km, at 70 %
for 11:00 UTC and 78 % for 23:00 UTC. Over this altitude range, relative humidity over ice in Fig. 7d increases slightly from 75 % to 80 % for both sondes. Unlike the 21 March 2017 example, the 11:00 UTC and 23:00 UTC sondes for 14 November 2017 differ significantly above 1.25 km. For the 11:00 UTC sonde, which corresponds to the times in plots 6b,c,d,e: As the temperature increases more swiftly, the relative humidity does so also, to to 90 % with respect to water, and to 107 % by 2 km with respect to ice. The relative humidity over water then continues to slowly increase to 94 % at 2.75 km, while the relative
humidity over ice decreases to slightly below 100 % by this altitude. An oscillating decrease is then seen in both relative humidity plots until 3 km, at which point relative humidity over both water and is about 80 %. After a short spike to higher relative humidity values just above 3 km, both profiles then decrease immediately to 20 % by 3.3 km. Above that point, relative humidity over water does not exceed 50 %, and relative humidity over ice does not exceed 45 %. The change from small to large gradients in altitude at 1.25 km is correlated with reaching the upper edge of the thicker cloud. Humidity remains high

as the sonde rises through the region with lower photon count returns, and then decreases very quickly as the top of the whole cloud is reached just after 3 km.

The 23:00 UTC sonde is somewhat different. In particular, the relative humidity values are as much as 10 % lower over water, and 20 % lower over ice, between 1.25 and 3 km altitude, never exceeding 85 % over water, nor 90 % over ice. The values are relatively constant, or slowly decreasing, up to slightly above 4 km, at which point a quick decrease in relative humidity is seen, which is similar to the decrease at 3 km in the 11:00 UTC sonde. Above 4.5 km, the response of the 11:00 UTC and 23:00 UTC sondes are similar. Neither the 11:00 UTC nor the 23:00 UTC profile is particularly smooth; there is lots of fine structure on the scales smaller than 100 m. Pursuing the humidity at higher resolution to match that of CRL may prove interesting, to see whether there is a correlation between the fine structure in the humidity and the laminations visible in the lidar data.

Figure 7e,f give the windspeed and direction for both sondes. The 11:00 UTC 14 November 2017 sonde is similar to the 23:00 UTC sonde from 21 March 2018: Wind direction is generally around 270° below 750 m, then shifts around toward 90° above that altitude. The wind direction in Fig. 7e is slightly more consistent at higher altitudes than it is in Fig. 5e. The windspeed stays below 9 m/s between the ground and 5 km, with a minimum at 750 m as the direction changes. The 23:00 UTC sonde on 14 November 2018 is quite similar to the 11:00 UTC sonde on that day with one wind direction below 750 m (including a bit more variability in direction for 23:00 UTC), and the opposite wind direction above that altitude. However, the directions in the 11:00 UTC and 23:00 UTC sondes are reversed with respect to each other. The windspeeds are also higher at 23:00 UTC, still generally increasing with altitude, but reaching up to 16 m/s. In the region of Figs. 7b,c,d,e, the 11:00 UTC wind profiles are relevant from 0.5 to 3.5 km. Examining Fig. 7b, a short (15 min) gap in the strong backscatter shown by the range-scaled photocounts is visible at 11:00 UTC between 1.25 km and 2.5 km, just at the time of the sonde. The laminations are far less pronounced during this gap than they are at other times of day. The sonde drifts in space during its ascent, and it takes some time for the sonde to rise. It is entirely possible that the sonde accesses some of this gap as well as other laminated parts of the cloud as it rises.

Light snow dominated the meteorological conditions reported at the ground for the first half of 14 November 2017. Hourly metorological observations recorded by the Eureka Weather Station on 14 November 2017 note snow at 00:00 UTC and 01:00 UTC, snow and blowing snow at 02:00 UTC, snow at 03:00 UTC, 04;00 UTC, and 05:00 UTC, ice crystals at 06:00 UTC, 07:00 UTC and 08:00 UTC, clear skies at 09:00 UTC, no reported condition at 10:00 UTC and 11:00 UTC, snow at 12:00 UTC through 15:00 UTC, no reported condition at 16:00 UTC and 17:00 UTC, clear skies at 18:00 UTC ,and ice crystals at 19:00 UTC, 20:00 UTC and 21:00 UTC, which are the final reports for the day. The temperature recorded at the weather station varied between -31.5° C and -38.9° C throughout the day.

## 5 Discussion

Before attributing the striped effect that we see in our data to geophysical phenomena, we apply due diligence to show that it is not an instrumental effect. Each of the topics covered by Sections 5.1- 5.4 address a specific instrumental or measurement ef-

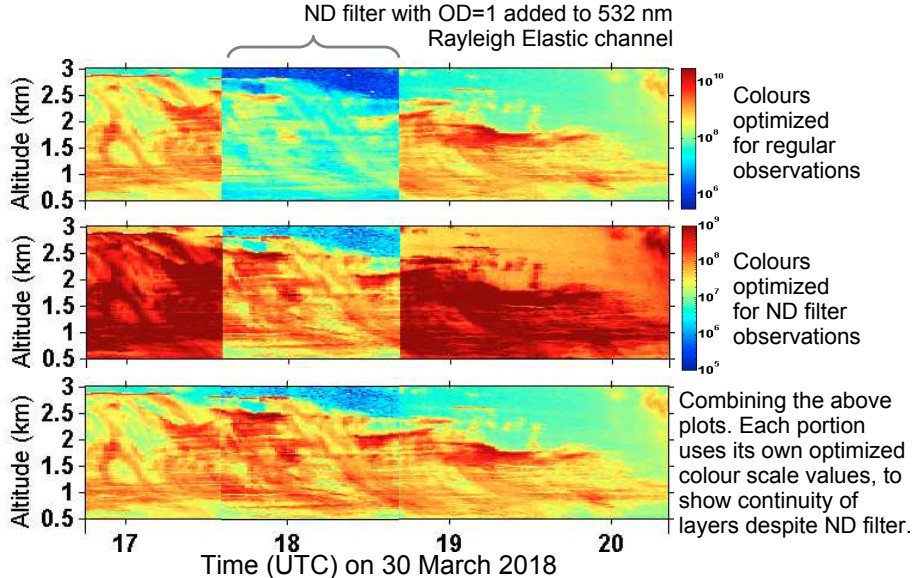

**Figure 8.** On 30 March 2018, during an event with the same features as previous examples, we placed an ND1 neutral density filter in front of the 532 nm Rayleigh elastic PMT for one hour. The stripes remained visible throughout the test. This is extra assurance that the PMT is not being saturated. The top panel has a colourbar which is optimized to show the stripes in the clouds before and after the ND filter test. The middle panel has a colourbar which is optimized to show the stripes during the ND filter test, when count rates were lower by a factor of about 10. The bottom panel is a combination of the first two plots. Measurements from all times during the test are shown at their own optimal colour scale so that individual layers may be identified and followed throughout the test.

fect/artifact which has been suggested by members of the broader lidar community as a possible indication that the laminations are not geophysical phenomena. Following that, we discuss some meteorological explanations for our observations.

## 5.1 Ruling out PMT saturation

As discussed briefly in Section 3, the analyses are made using glued count rate profiles, which make use of photon counting
5 signals in regions where the photon count rates are linear, and equivalent analogue signals in any region for which the photon counting rates become nonlinear. During routine processing, all regions in which the analogue signals meet or exceed the counting limits of the analogue-to-digital converter are excluded from the retrieved profiles. For all measurements in this manuscript, the PMTs were not being operated near their maximum analogue count rates, so the likelihood of the laminations being PMT saturation artifacts is low.
10    Further, any saturation effects should serve to smooth out the profiles at high count rates, rather than inducing the oscillating count rates as we observe as the laminated cloud phenomena. In order to clearly demonstrate that these laminated features persist at much lower photon count rates, we performed a measurement with the aid of neutral density filters to lower the signal levels.

During a 30 March 2018 event which exhibited the type of layers discussed in this paper, we placed a neutral density (ND) filter with optical density 1 (ND1) in front of the 532 nm Rayleigh elastic channel's PMT. This reduces all count rates entering the PMT by a factor of 10. The ND filter was left in place for one hour, and then was removed. The results of this test are given in Fig. 8. It is clear from the composite plot in the bottom panel of Fig. 8 that the layers seen in the clouds during regular measurements (before 17:40 UTC and after 18:40 UTC) are continuous throughout the time that the ND1 filter is in place (17:40 UTC to 18:40 UTC). Since the layers are still seen at count rates which are lower by a factor of 10 compared to regular observations, we conclude that PMT saturation is not the cause of the layers.

## 5.2 Ruling out PMT ringing

PMT ringing effects induced by the "nonlinear response of a detector-amplifier combination to a signal larger than the dynamic range of the combination" Kovalev and Eichinger (2004) can produce phenomena in the backscattered lidar data which could, to first order, be described in a similar to the laminations seen in this manuscript: namely, vertical structures on the scale described in the manuscript. However, (a) we would not expect to see PMT ringing if the PMT is not being saturated (this has been excluded as covered in Section (5.1), above), and (b) we would expect the effects to be different than what we see in the cloud measurements.

There are several important differences between the expected ringing PMT response and the response from geophysical laminations within the clouds themselves. For a visual example of the ringing phenomenon, see Kovalev and Eichinger (2004), Figure 4.6. There, we see the "periodic nature of the returns above the cloud layer". Those backscattered returns are nearly precise repeats of the same shape as the cloud signals below. There is regular repetition as a function of range from the lidar.

On the converse, in the CRL measurements of the laminations (e.g. Figs 1, 4a, and 6a), the layers are not actually a regularly repeating pattern. They come and go, merge into one another, change in vertical extent with altitude, and are not regularly spaced in range from the lidar by any obvious geometric function.

Therefore, we do not interpret the laminated cloud features to be effects of PMT ringing.

## 5.3 Ruling out laser power fluctuations

Laser power fluctuations would induce increases and decreases to the range-scaled photocounts values as a function of profile number throughout the day (i.e. would manifest as vertical stripes in the plot), and cannot produce the layered features that we see in the figures, which are a function of altitude (and appear therefore as horizontal stripes in the plot).

## 5.4 Ruling out timing and electronics systematics

If the layers were a result of a timing offset, constant electronic noise, or similar, we might first expect the layers to be truly constant in altitude. They are not. The layers drift slowly up and down, split apart and recombine, are not always at the same altitudes, and do not always have the same individual layer thickness. Therefore we find systematic timing and electronics issues to be an unlikely source for the features displayed in the plots.

## 5.5 Meteorological considerations

The analysis of our measurements leads us to interpret the layered features as geophysical. Thus, the stripes in the plots are interpreted to be fine laminations within the cloud. We see these features in several types of meteorological conditions, and have seen evidence of them in more than 3 years of lidar measurements. We see them at various times of year.

Several conditions which currently seem to be associated with the laminations, and which must be taken into account when suggesting meteorological explanations, are:

1. Association with thermal/convective stability:

   The winter examples shown here exhibit a strong temperature inversion, and the summer example also has a stable temperature structure. Not all of the laminations are confined to the altitudes covered by the temperature inversion, when present.

   Radke et al. (1989) suggest, based on the work of Andraea et al. (1988), McElroy and Smith (1986) and Wakimoto and McElroy (1986), that thin, elevated, hazes can occur also at mid-latitudes and these, too, occur only in regions of great thermal stability. If the atmosphere were not vertically stable, then these laminations could not persist as they would be removed by the vertical mixing. Perhaps this is a necessary condition for such laminations. An indication to the contrary is Hobbs and Rangno (2008), which has found cloud features similar to CRL's laminated cloud layers in regions of both thermal stability and thermal instability - often within a single cloud. It is possible that the laminations form in a stable region and then drift outside that region, persisting for some time before being obliterated by vertical motions.

   Our explanations here must be consistent with stable thermal profiles, although there may exist cases of similar laminations arising in other situations.

2. Association with precipitation:

   Each case of laminated clouds shown in this paper exhibited fall streaks within the cloud, and precipitation to the ground.

   We will carry out a detailed search for cases of these laminations which are and are not associated with precipitation events at the ground. It is as yet unclear whether precipitation is a necessary condition for, and/or obligatory result of, these laminations.

   Explanations must allow for precipitation to the ground, since it happens in the cases shown here.

3. Association of regions of high/low range-scaled photocount rates with regions of low/high depolarization parameter:

   There are regions in all plots with depolarization parameter $d < 0.1$, which indicates clear air, liquid (quasi-spherical) droplets, horizontally-oriented ice plates, or specific types of aerosols. For those $d < 0.1$ regions in which the range-scaled photocounts are high, clear air is unlikely to be the scatterer responsible; liquid droplets, oriented ice particles, and/or aerosols are more likely. Thus, our explanations must allow for the creation of, or continued existance of (if created elsewhere), liquid droplets, ice, and/or low-depolarizing aerosols.

There are certain regions in which the range-scaled photocount plots display laminations, but which are homogenous in terms of depolarization. Examples include 0.6 km to 1 km from 10:50 UTC to 11:10 UTC on 14 November 2017, and 22:00 UTC to 24:00 UTC from 1.8 km to 2.25 km on 21 March 2017.

Similarly, there are regions in which the laminations in the range-scaled count plots are less pronounced and/or absent, interrupting the consistent layered structure of the rest of the cloud. Such locations tend to have high depolarization parameter associated with high range-scaled count rates (e.g. the diagonal feature descending from 1.5 to 0.5 km from 0930 UTC to 09:50 UTC on 14 November 2017, or the smaller patch on that same day at 11:10 UTC from 0.5 to 0.6 km). Precipitating frozen particles would be consistent with this observation, and thus must not be considered to be impossible in our hypotheses.

## 5.6 Suggested explanations for the laminated phenomena

At this time we do not have a complete explanation of these measurements. Several explanations may be consistent with the results. Hypotheses currently under consideration include interactions with a background field of (possibly layered) aerosols, preferential condensation and/or precipitation and/or evaporation of particles, and tropospheric waves.

### 5.6.1 Preferential condensation and/or precipitation of particles via interaction with background field of aerosols

If we begin with a background of aerosols, perhaps already layered, in a relatively humid atmosphere, these aerosols would allow the moisture to condense upon them. Regions with more aerosols (e.g. in an aerosol layer) would be likely condensation sites, and regions with fewer aerosols (e.g. between the layers of aerosols) would not. Any existing condensed particle will scavenge remaining moisture preferentially, with larger particles growing at the expense of the nearby smaller ones and any moisture.

The larger particles at some size would become large enough to fall and precipitate out of the cloud. Any small regularity in spacing between populations of particle sizes will be exacerbated into stripes such as those we see in the presence of a very stable atmosphere - there is no vertical mixing to disrupt the pattern. The precipitating particles should be able to fall out of the cloud without disrupting the overall layered structure.

The particles may freeze immediately upon condensation, or during precipitation. If the particles which precipitate out are frozen, and those remaining in the cloud are liquid, this would show up as bright layers with low depolarization (liquid droplets which have not yet precipitated out), and in-between regions of low brightness and higher depolarization where the few frozen particles that remain are still growing.

Many other condensation/precipitation processes are possible in this mixed-phase environment, and all should be considered when we carry out a thorough analysis. In the above situations, condensation is occurring preferentially in certain layers, and precipitation exacerbates the laminated situation.

A question then remains: What would cause the background of aerosols to have any amount of layered structure in the first place? Perhaps the explanations of Radke et al. (1984) and Radke et al. (1989) would apply. One suggested cause for morphologically-similar haze laminations is "the advection of thin hazy regions into the generally clean polar airmass and by

the extreme thermal stability of the lower troposphere" (Radke et al., 1989). In Radke et al. (1984), strong windshear between haze layers and clear layers leads the authors to conclude that polluted layers of air are advected into regions of clean air, rather than a complete unit of haze layers interspersed with clean air being advected together into the region of their measurements. Further investigation of wind shear and detailed temperature structure at Eureka will be beneficial for testing this hypothesis at CRL.

### 5.6.2   Inhomogeneous evaporation from a uniformly condensed field

It is also possible to begin with a uniformly condensed field, and allow evaporation alone to then carry on in a non-uniform manner, leading to preferentially dried sections of the cloud. Holographic imaging results from Beals et al. (2015) demonstrate that turbulent clouds which begin with homogenous features in terms of cloud droplet number density and droplet size can become quite inhomogeneous as mixing occurs, leading to filamented structures at centimetre scales. As mixing carries on, they do not see evidence of droplets evaporating uniformly across the population (constant number density, decreasing droplet size), but instead see certain droplets evaporating entirely while the remaining droplets retain their original size (decreasing number density, constant droplet size). This mechanism could be at play within the clouds seen by CRL as well. In lidar measurements we cannot separate the droplet size from number density, as we measure a quantity proportional to a combination of these variables. Nonetheless, our observations are consistent with those of Beals et al. (2015): Regions with high number density $\times$ cross sectional droplet size, interspersed vertically with regions having low values of that quantity (and indeed being perhaps nearly free of droplets). Precipitation would still need to be accounted for (as per Section 5.5) in this scenario.

### 5.6.3   Persistence of layers

Persistence of the laminated cloud features measured by CRL may find its explanation in Mahrt (2014), which cites Sukoriansky and Galperin (2013): in the case of strong stratification, material can be transferred more effectively by horizontal diffusion than by vertical diffusion. This would serve to preserve any material within its own horizontal layer, rather than spreading it out to the relatively emptier regions between the initial layers. Mahrt (2014) is focused on the boundary layer, but the turbulence theory cited applies equally to other areas of the atmosphere. Therefore the laminations in the Eureka clouds may persist due to the so-called two-dimensional (quasi-horizontal) modes which are not significantly coupled in the vertical direction. Mahrt (2014) indicates that these modes can be transient, reforming and breaking down, over a variety of time scales.

### 5.6.4   Tropospheric waves

This hypothesis is that the laminations are the effect of gravity waves or other waves travelling through the troposphere. The waves would have a vertical wavelength of approximately 15 m (for each bright layer to be 7.5 m, and each in-between layer to also be 7.5 m). The atmosphere can sustain gravity wave motions so long as the density is decreasing with altitude. That is to say, the environmental lapse rate must be stable. Regions of adiabatic cooling will have water vapour more readily able to condense onto condensation nuclei, while the regions of adiabatic heating will be less likely to do so. Those particles which

do exhibit condensation can then scavenge any surrounding moisture, as previously described, and precipitate out of the cloud. This explanation is also consistent with the layered features seeming to cease by 4 to 5 km altitude: Gravity waves are unable to continue propagating through unstable regions, which may occur at higher altitudes. Likewise, they will deposit their energy anywhere that the horizontal phase speed matches the speed of the background winds. In general, horizontal wavelengths of gravity waves tend to be much larger than the vertical wavelengths of those same waves; hence, we see finely laminated structure in the vertical (Hocking, 2001).

### 5.6.5 Horizontal spatial distribution of clouds

With the combining and separating of certain layers, there is also the possibility that these layers are the effect of a projection of a horizontally patchy cloud onto a measurement which is extended in time. One minute (CRL's maximum time resolution) is a long time to be watching clouds drift by if windspeeds are high, and the laser beam subtends a horizontal circle of approximate diameter 7.5 m by 5 km altitude. Further, we are interpreting a 3-D atmosphere with a 2-D measurement. Perhaps these are not contiguous layers, but rather are multiple patchy laminated clouds. With the present plots, we cannot distinguish the two situations. Understanding the horizontal spatial distribution of the observed clouds will be helpful.

### 5.6.6 Discussion summary

Further investigations are requisite in order to rule in or out any of the hypotheses above, or other, possibilities. Further analysis with CRL's other channels, and Eureka's other instruments, will surely narrow down the possibilities. At the moment, we have a very intriguing phenomenon which appears to occur in frequent events at our lidar and we continue to add ancillary measurements for our next campaign.

## 6 Conclusions

Measurements of range-scaled photocounts at 532 nm and 355 nm, photocount ratios 532/355 nm, and 532 nm linear depolarization parameter from the CRL at Eureka, Nunavut have detected numerous instances of finely laminated cloud structures during all times of year. The individual laminations are measured to be as thin as 7.5 m per layer, with thinner features not being resolvable by CRL.

Generally, layers with high range-scaled photocount rates are associated with layers of low depolarization parameter values. Occasionally, the layered structure is interrupted by homogenous regions in terms of both range-scaled photocounts and depolarization.

The laminated clouds have, to date, only been measured during periods of precipitation reported at the ground: rain and snow. They also, for examples studied to date, seem to be associated with a stable thermal troposphere, including but not limited to days with strong temperature inversions.

This paper provides the motivation for further analysis of data sets from CRL and other high-vertical-resolution tropospheric lidars, particularly those in polar regions. The laminated cloud structures presented here are evidence that the mixed-phase

clouds at Eureka are frequently not homogenous, and should not be treated as such during investigations of condensation, precipitation, and other internal microphysical processes. While the contribution of such clouds to the regional radiation budget may be precisely equal to that of homogenous clouds having the same average optical properties, it does not necessarily follow that their internal processes are identical.

Further work will be done to combine these high-resolution CRL measurement products with both low-resolution more sophisticated CRL measurement products and with high resolution measurements from other instruments at Eureka. The combination of these efforts will lead to better hypotheses and explanations for the 7.5 nm-scale features which we now know to be frequently present in Arctic clouds at Eureka.

## 7 Data availability

Data used in this paper available upon request from corresponding author (emccull2@uwo.ca).

## Appendix A: Depolarization Uncertainty

For completeness, depolarization uncertainties for the two main dates examined in this paper are presented here. Figure 9 for 21 March 2017, and Fig. 10 for 14 November 2017.

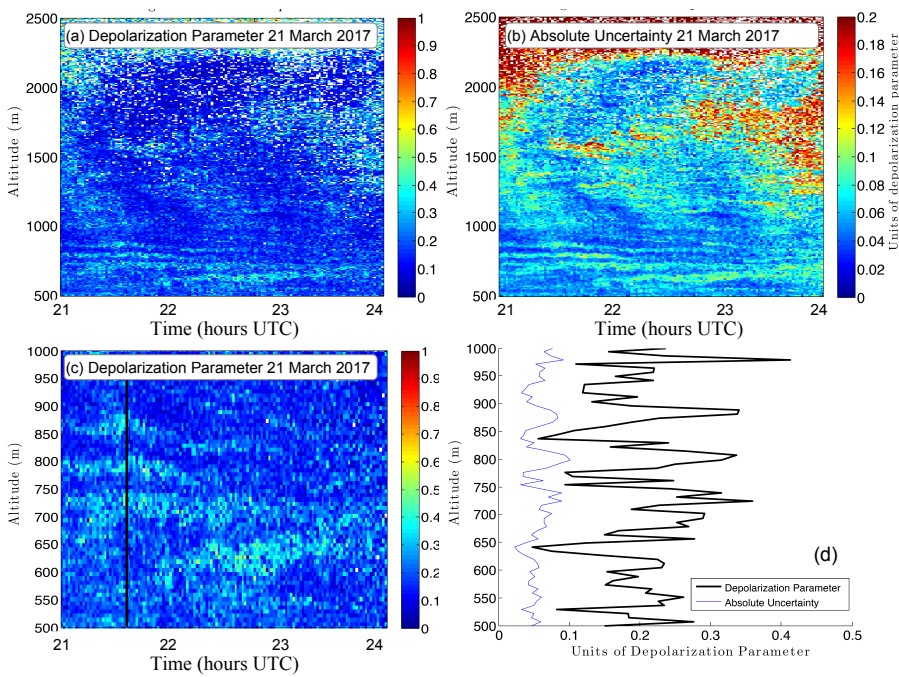

**Figure 9.** 21 March 2017. **(a)** is the depolarization parameter from 500 to 2500 m on a colour scale between 0 and 1. **(b)** is the corresponding absolute uncertainty in units of depolarization parameter, on a colour scale between 0 and 0.2. **(c)** is an enlarged portion of **(a)**. Black vertical line indicates the profile plotted in **(d)**. **(d)** shows a single one-minute profile of depolarization parameter in black, and its corresponding absolute uncertainty profile in blue.

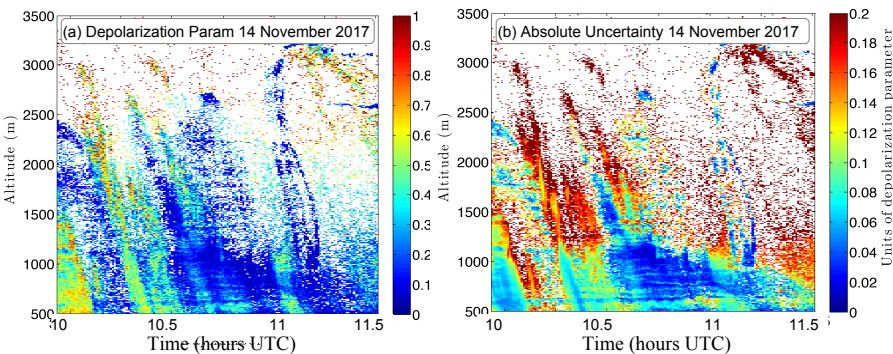

**Figure 10.** 14 November 2017. **(a,b)** same format as 9**(a,b)**.

## Appendix B: Calculations of RH over ice

5 Relative humidity with respect to liquid water ($RH_w$) is converted to relative humidity with respect to ice ($RH_i$) using the Goff-Gratch formulations for saturation vapour pressure (Goff and Gratch (1946), in List (1949)). Saturation vapour pressure over water, $e_w$, can be calculated via equation B1:

$$\log_{10} e_w = -7.90298\left(\frac{T_s}{T} - 1\right) + 5.02808\log_{10}\left(\frac{T_s}{T}\right) - (1.3816 \times 10^{-7})(10^{11.344\left(1 - \frac{T}{T_s}\right)} - 1)$$

$$+ (8.1328 \times 10^{-3})(10^{-3.49149\left(\frac{T_s}{T} - 1\right)} - 1) + \log_{10} e_{ws}, \tag{B1}$$

10 in which $T$ is the radiosonde temperature in Kelvin, $T_s = 373.16\,\text{K}$ is the steam point temperature of liquid water, and $e_{ws} = 1013.246\,\text{mb}$ is the saturation pressure of liquid water at the steam point temperature (at 1 standard atmosphere). Saturation vapour pressure over ice, $e_i$, can be calculated via equation B2:

$$\log_{10} e_i = -9.09718\left(\frac{T_o}{T} - 1\right) - 3.56654\log_{10}\left(\frac{T_o}{T}\right) + 0.876793\left(1 - \frac{T}{T_o}\right) + \log_{10}(e_{io}), \tag{B2}$$

in which $T_o = 273.16\,\text{K}$ is the ice point temperature, and $e_{io} = 6.1071\,\text{mb}$ is the saturation pressure of ice at the ice-point temperature (at 0.0060273 standard atmospheres). Relative humidity with respect to ice, in percent, is then equation B3:

$$RH_i = \left(\frac{e_w}{e_{io}}\right)RH_w \tag{B3}$$

## Appendix C: A summer example of layers on 26 August 2017

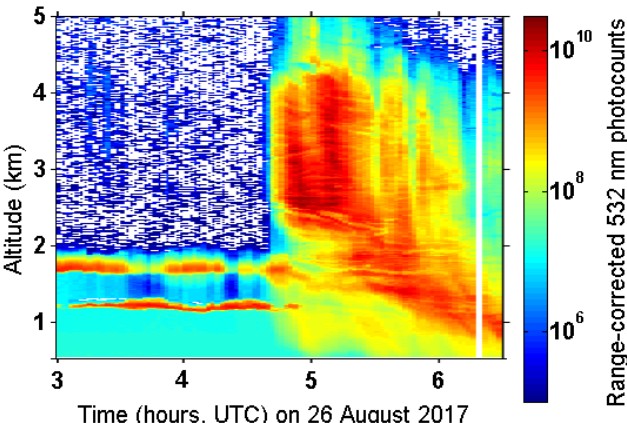

**Figure 11.** A summer example of fine-scale structure in clouds at Eureka. Plot of 532 nm range-scaled photocounts from 26 August 2017 **(a)**. The layers are most visible after 05:00 UTC. The measurement was stopped due to rain at 06:30 UTC.

The layering seen in the Arctic clouds above CRL are not only seen during cold times of year, as in the 21 March and 14 November examples. They are also occasionally seen in summer, such as 26 August 2017. Before 04:45 UTC Fig. 11

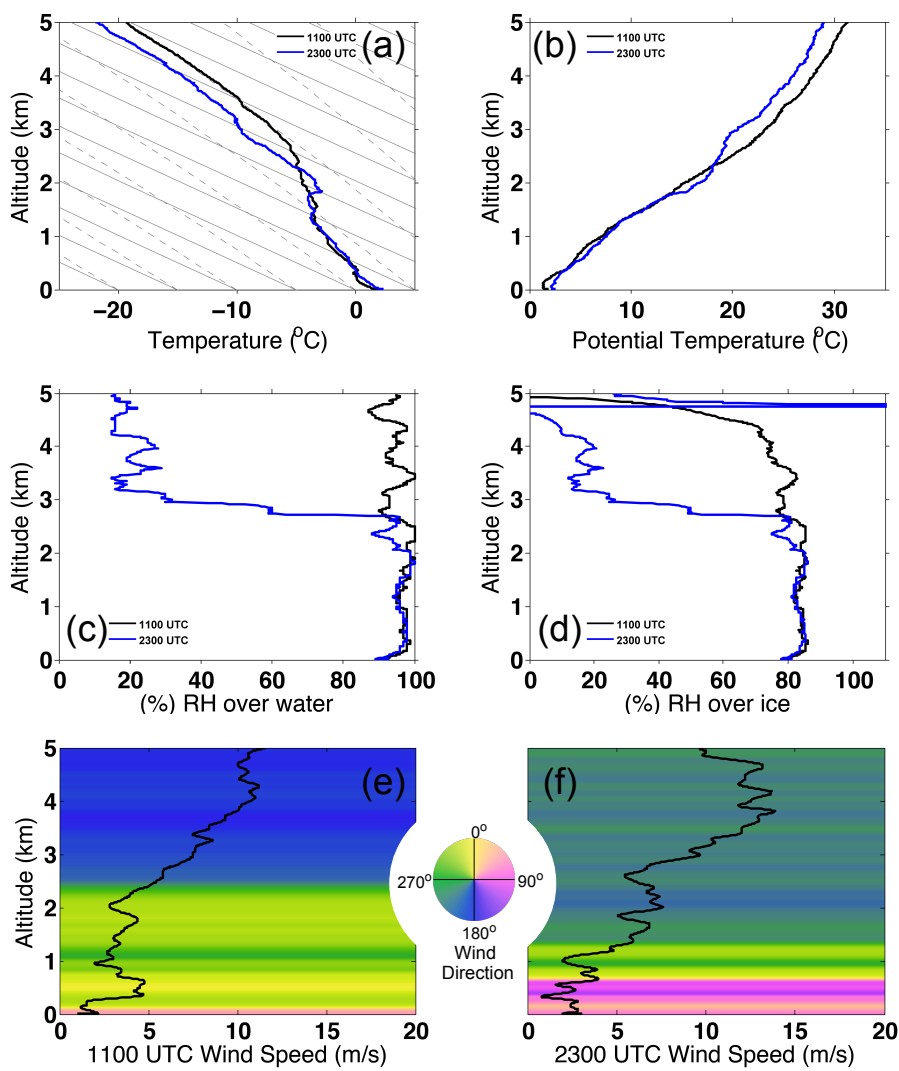

**Figure 12.** 26 August 2017. From the two daily rasiosondes launched by the Eureka Weather Station: **(a)** is the temperature; grey solid lines are dry adiabats, and grey dashed lines are moist adiabats. **(b)** is the potential temperature. **(c,d)** are the relative humidity with respect to water and ice respectively. **(e,f)** are the wind speed (black line) and direction (coloured background).

shows optically thick low-lying clouds which are typical of summer in Eureka. Because the lidar is largely extinguished by

10   these low clouds, we cannot discern details of any clouds above that altitude. There does appear to be some increase in signal between 3 and 4 km from 03:15 - 03:30 UTC, so it is highly likely that there are much thicker clouds above the low ones. After 04:45 UTC, we can see the full extent of some clouds which range from 0.5 to 4.5 km altitude. The same layering is present in these vertically extended clouds as we have seen in the previous examples in this paper.

26 August 2017 began with the lidar closed due to rain. Measurements were possible from 00:30 - 06:30 UTC. Despite rain

being reported at the Eureka Weather Station in the hourly meteorological observations, there was so little during this time as to not impede measurements. At 06:30 UTC, the rain again became heavy enough that measurements ceased. The 355 nm laser was not operating during this measurement, so a full investigation of this case will not be presented here.

The depolarization measurements (not shown) indicate that the high-backscatter parts of the clouds before 05:00 UTC (red in Fig. 11) have low depolarization parameter values of about 0.1, and that after 05:00 UTC the regions shown in yellow below

2 km in Fig. 11 have higher depolarization parameter of about 0.6. The interpretation is that the highly attenuating clouds early in the day are liquid, and that the precipitation which begins at 05:00 UTC consists of frozen particles. There is insufficient sensitivity in the preliminary depolarization product to determine the depolarization parameter within the layered region of the cloud after 05:00 UTC.

Hourly metorological observations recorded by the Eureka Weather Station on 26 August 2017 note cloudy conditions at

00:00 UTC, rain at 01:00 UTC, rain and fog at 02:00 UTC through 05:00 UTC, rain, snow showers and fog at 06:00 UTC, rain and snow showers at 07:00 UTC, and reports of rain and fog for the remainder of the day. The temperature recorded at the weather station varied between $0.8^\circ$ C and $2.9^\circ$ C throughout the day.

Figure 12a,b give the radiosonde temperature and potential temperature profiles, and Fig. 12c,d the radiosonde relative humidity profiles with respect to liquid water and with respect to ice. The temperature profiles were very similar at 11:00 UTC

and 23:00 UTC, but the relative humidity measurements differ drastically above 2.7 km. As neither sonde was launched during the CRL measurement period, we cannot draw strong conclusions from these. Still, the adiabats plotted in 12a provide a point of comparison for the temperature profiles in terms of thermal stability: On 26 August 2017, as for the other dates shown in this paper, the atmosphere was relatively stable in the region of the cloud laminations. Figure 12e,f give the windspeed and direction for both sondes.

*Author contributions.* E. M. McCullough: Operation and maintenance of the lidar. Data analysis. Writing of analysis MATLAB code. Manuscript preparation. J. R. Drummond: Principal Investigator of PEARL laboratory. Contribution to manuscript preparation. T. J. Duck:

5   Development of the CRL laboratory, and initial Principal Investigator of CRL lidar. Contribution to manuscript preparation.

*Competing interests.* The authors declare that they have no conflict of interest.

*Acknowledgements.* This research is currently supported by the Natural Sciences and Engineering Research Council, Environment and Climate Change Canada and the Canadian Space Agency.

PEARL has been supported by a large number of agencies whose support is gratefully acknowledged: The Canadian Foundation for Innovation; the Ontario Innovation Trust; the (Ontario) Ministry of Research and Innovation; the Nova Scotia Research and Innovation Trust; the Natural Sciences and Engineering Research Council; the Canadian Foundation for Climate and Atmospheric Science; Environment and Climate Change Canada (ECCC; who also provided the radiosonde data); Polar Continental Shelf Project; the Department of Indigenous and Northern Affairs Canada; and the Canadian Space Agency. This work was carried out during the Canadian Arctic ACE/OSIRIS Validation Campaigns of 2016 and 2017, which are funded by the Canadian Space Agency, Environment and Climate Change Canada, the Natural Sciences and Engineering Research Council of Canada, and the Northern Scientific Training Program. This particular project has also been supported by NSERC Discovery Grants and Northern Supplement Grants held by James R. Drummond, Robert J. Sica, and Kaley A. Walker, and the NSERC CREATE Training Program in Arctic Atmospheric Science (PI: Kim Strong).

In addition, the authors thank the following groups and individuals for their support during field campaigns at Eureka: PEARL site manager Pierre Fogal; Canadian Arctic ACE/OSIRIS Validation Campaign project lead Kaley A. Walker, CRL operators: Graeme Nott, Chris Perro, Colin P. Thackray, Jason Hopper, Shayamila Mahagammulla Gamage, and Jon Doyle; Canadian Network for the Detection of Atmospheric Change (CANDAC) operators: Mike Maurice, Peter McGovern, John Gallagher, Alexei Khmel, Paul Leowen, Ashley Harrett, Keith MacQuarrie, Oleg Mikhailov, and Matt Okraszewski; and the Eureka Weather Station staff. Thanks to Jean-Pierre Blanchet and other members of CANDAC for their helpful discussions.

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
