# Peer review of "Lidar measurements of thin laminations within Arctic clouds"

_Atmospheric Chemistry and Physics, 2018_

## Referee Comment (RC1) · Anonymous Referee #2 · 14 Oct 2018

This paper describes a very interesting set of observations of persistent fine-scale vertical laminations within Arctic clouds. The measurements are intriguing and the authors have conducted convincing tests to show that the laminations are not instrumental artifacts. The paper is largely descriptive: the authors describe the observations and the conditions under which the laminations have been observed, and briefly describe other observations of laminated aerosol and cloud structures. They offer little in the way of explanation for the observed phenomena, however, which seems to me a major shortcoming that should be rectified before publication in Atmospheric Chemistry and Physics.

A few specific comments:

The figures showing range-scaled photocounts on log scales are a little hard to inter-

pret. How deep are the laminations/striations? Are they closer to 10% or 90% of the total backscatter? More quantitative information would help the reader consider the possible roles of cloud vs interstitial aerosol particles.

To first order, the laminations are reminiscent of the fog striations seen in cold pools under near stable conditions (Stably Stratified Atmospheric Boundary Layers, L. Mahrt, Annual Review of Fluid Mechanics 2014 46:1, 23-45). What are the wind conditions here? Wind profiles and Richardson numbers would be a useful addition, and potential temperature profiles would also be more instructive that the included temperature profiles.

---

## Referee Comment (RC2) · Anonymous Referee #1 · 19 Oct 2018

The submitted manuscript provides observational evidence that thin cloud layers may exist in the Arctic troposphere given certain atmospheric stability conditions. This manuscript demonstrates that high range resolution measurements are possibly needed to capture full cloud processes using lidar observations well in excess of previously considered range resolution. The manuscript is well written, well sourced, and provides, to my knowledge, previously unreported data of scientific interest. However, this manuscript is also speculative in nature providing few concrete explanations for observational phenomena. The conclusion calls for work that should, in my opinion be in this work and can not be removed from the scope. It is my recommendation that this manuscript should be accepted with major revisions (really major additions) with a more complete analysis of the critical data accumulated.

[Figure]

Major Comments:

1) Figure 2: There appears to be very little attenuation of your light within this cloud. This raises concern for me about multiple scattering enhancing your signal. The Nott et al. 2012 paper described the field of view of the system at 0.3-2mrad. What were you running for this data collection? Is multiple scattering a concern? 3d multiple scattering effects could be very difficult to separate from physical structure and could add (or smooth out) structure on the order of a few range bins depending on the physical features of the cloud.

2) You say several times that taking data at lower resolution would cause the thin features to be covered (example on Page 2, Line 19-20). I am skeptical that this would completely remove some of the features you see, though I do not doubt it will change them. For example, the thick count layer at 3km from 4-5.5 UTC in Figure 1 would possibly remain. I believe you should show high vs. degraded resolution to better illustrate this point. Further, it will allow you to quantitatively assess, both what other investigators should be looking for in their lower resolution data and define to what extent data is masked. Specifically, it would help place your work in the context of the previous authors you describe on Page 2, Lines 28-33. Additionally, it will suggest how fruitful further analysis might be, combining data with the low-resolution lidar data products.

3) Why do you not apply overlap corrections? Showing data below 500 meters and not overlap correcting is confusing to me and a bit misleading in places. Suggest either applying the corrections or removing all data below full overlap for clarity.

4) Page 10, Lines 27-29 and Figure 3 and Figure 4: Depolarization contours are very noisy. I would argue they are almost unhelpful. In fact, given the results of the McCullough et al. (2018) paper, I am questioning if you have the sensitivity in the depolarization channel to make the described measurements at 1 minute resolution. At the very least, contours of depolarization error bounds should be shown to inform your reader how far they can trust the interpretation of depolarization.

5) Section 4.3: Perhaps this is best used as an appendix. It is less convincing than the other 2 cases based on the level of information you are able to provide. It might be more helpful to summarize your measurements to describe the percent of time you see clouds with such vertical laminations.

6) Section 5.1-5.3: The discussion in these 3 sections is a major weakness of the paper in my opinion. I do not find the discussion particularly convincing because the topics discussed, while likely being familiar to a reader knowledgeable of lidar hardware, is not particularly well constructed in my opinion. My concerns are as follows: a) PMT or saturation more generally should serve to smooth your profiles in every case I can imagine. If the section of your glued profile originates from photon counting data, photons will be under-reported and thus thick clouds will seem thinner. If the portion of the profile is from the analog counting system and you are under reporting intensity (or even clipping the ADC), you are operating so far outside of the designed regime of the detectors that the data is likely not valid. Additionally, you claim to have corrected it in Section 3. b) Signal induced noise should be slow (microsecond time scale) and extensive in altitude. c) PMT ringing on the other hand is something I would think could cause vertical structure on the scale described. I would think this is the major instrument effect to investigate. d) I agree with your conclusion about laser power fluctuations. So much so that I would likely not even mention it in this analysis. e) I agree mostly with your timing electronics conclusions but if you have an issue, it might not be stable in altitude. If you have 2 or more different clock speeds (from triggering, seeding, q switching, or your counting system clock drifting slightly), you could possibly alias one rate onto the other making your observations move in altitude. That would likely be a systematic shift observable at all altitudes though, and as such easy to identify.

7) I am surprised that the authors have not included lidar data that could be very helpful. They do call for more analysis in the conclusion. That said, without this analysis, I am not convinced that this work is a major observational contribution. Some omissions that

I believe should be seriously considered (at minimum) are: a) I find myself surprised that the authors use radiosondes and not rotational Raman measured temperatures and vibrational Raman measured water vapor. This is especially true of Figure 4 where the thermodynamic structure changes dramatically over the observation period. The data need not be at 1 min resolution to be helpful. b) I also find myself surprised that basic summary statistics of occurrence frequency or bounding relative humidity or temperature are not provided. At minimum, I would expect to see some observational bounds on conditions described in Section 5.4. c) I am not sure raw photon counts are sufficient to quantitatively show the structures within clouds. Calibrated backscatter coefficients would be much more useful. Additionally, they remove uncertainty sources such as laser power fluctuations.

Minor Comments:

1) Page 1, Line 4: It obviously depends on your target but 1 min time resolution might not be particularly high resolution. Suggest dropping the word "high" here. Also on Page 5, Line 10

2) Figure 2: At 3km, the range correction should be $9 \times 10^6$. The counts that you are showing are therefore on the order of 10-100. Is that correct? If so, counting statistics worry me. It is impossible to tell here what wiggles are due to scattering phenomena and what wiggles are due to counting statistics. Suggest adding error bars to clarify.

3) Page 3, Line 1-8: The following paper and references therein may be of interest to the authors as motivation for cloud structure size scales: Beals, et al., "Holographic measurements of inhomogeneous cloud mixing at the centimeter scale," Science 350, 87–90 (2015).

4) Page 4, Line 17: Referring to a broad class of elastic scatter lidars as Mie lidars is very imprecise. Suggest modifying to "elastic scatter" as you have no way of verifying that all scatterers are spheres.

5) Page 6, Line 26: This sentence is confusing because your lidar counting system has already binned single photon data to 7.5 meters and 1 minute. Suggest modifying this sentence to something like: "No further binning was performed. "

6) Figures 3 and 4: I believe there are several ways to calculate relative humidity with respect to ice. There are several parameterized versions or more simple versions. They do not all result in identical values given identical inputs. Suggest adding a citation to describe the method you use.

7) Figure 4 Caption: Suggest shortening by describing panels a-f as "same as Figure 3" or similar.

8) Page 9, Lines 27-29: Low depolarization is consistent with observations of preferentially oriented ice crystals. Suggest clarifying that high depolarization is "…inconsistent with interpretation as randomly oriented ice particles." Note that the following might be of interest as well, especially Appendix A: Silber, et al., "Polar liquid cloud base detection algorithms for high spectral resolution or micropulse lidar data," J. Geophys. Res.: Atmos. (2018).

9) Page 10, Line 15 and elsewhere: I find the use of numbers like $1\times10^{10.5}$ to be difficult to interpret. Suggest changing to integer powers: $1\times10^{10.5}=3\times10^{10}$ or much less preferably changing to dB.

10) Page 11, Line 16-17 and throughout the manuscript: I assume your sondes are reporting their raw data with respect to water. Are you reporting all relative humidity values with respect to ice? It is clear in the figures but less so in the text. Suggest adding "w.r.t ice" or "w.r.t. water" throughout the text to clarify or inserting a blanket statement specifying how all data are reported.

Technical Corrections:

1) Page 10, Line 5: "…the air is remains…"

2) Page 10, Line 14 and elsewhere: "The clouds[,] which contain…" The use of the

word "which" requires use of a comma in most places.

3) Page 19, Line 18: I believe the paper you refer to here is in the January 2012 publication, not 2011.

---

## Author Comment (AC1) · 29 Dec 2018

**Referee Comment 1: The authors describe the observations and the conditions under which the laminations have been observed, and briefly describe other observations of laminated aerosol and cloud structures. They offer little in the way of explanation for the observed phenomena, however, which seems to me a major shortcoming that should be rectified before publication in Atmospheric Chemistry and Physics.**

**Author response:**

Some proposed explanations for the observed cloud laminations have now been added to the manuscript.

**Change to manuscript:**

We have added a new Section 5.6 to the manuscript: "Suggested explanations for the laminated phenomena".

New references:

[Beals2015CloudHolography]
Beals, M. J., Fugal, J. P., Shaw, R. A., Lu, J., Spuler, S. M., and Stith, J. L.: Holographic measurements of inhomogeneous cloud mixing at the centimeter scale, Science, 350, 87 − 90, 2015.

[Hocking2001GravityWavesWebsite]
Hocking, W. K.: Buoyancy (gravity) waves in the atmosphere, http://www.physics.uwo.ca/~whocking/p103/grav_wav.html, 2001.

[Mahrt2014StablyStratBoundaryLayers]
Mahrt, L.: Stably Stratified Atmospheric Boundary Layers, Annual Review of Fluid Mechanics, 46, 23–45, 2014

**Referee Comment 2: The figures showing range-scaled photocounts on log scales are a little hard to interpret. How deep are the laminations/striations? Are they closer to 10% or 90% of the total backscatter? More quantitative information would help the reader consider the possible roles of cloud vs interstitial aerosol particles.**

**Author response:**

Examining Figure 2, blue curve, gives a few calculable examples. Let's consider by how many percent of the range-scaled photocounts the in-between layers (yellow in Fig 1) drop the signal compared to the values in the layers themselves (red/orange in Fig 1):

One of the "deeper" laminations gives a result of $((10^{9.316} - 10^{8.833})/(10^{9.316})) \times 100\% = 67.11\%$, while the shallower laminations produce results such as $((10^{9.312} - 10^{9.103})/(10^{9.312})) \times 100\% = 38.2\%$.

These are fairly representative values. Therefore the range-corrected signal drops by between about

35% and 70% of maximum local value between layers.

**Change to manuscript:**

The text in bold is added to the end of the paragraph on Page 2 lines 15-18 : ``Figure 2 shows selected profiles of range-scaled 532 nm photocounts from Fig. 1 as a function of altitude for four consecutive minutes just after 06:40 UTC, each offset by $1 \times 10^{0.6}$ along the x-axis, between the altitudes of 3 to 4 km. There are clearly horizontal coherent structures in the cloud in space (aliased to time by motion over the lidar) at least down to the 7.5m height resolution of the lidar. **The regions between the laminations generally exhibit range-scaled signals between 35 and 70 % lower than the signals in the laminations immediately above and below.''**

**Referee Comment 3: To first order, the laminations are reminiscent of the fog striations seen in cold pools under near stable conditions (Stably Stratified Atmospheric Boundary Layers, L. Mahrt, Annual Review of Fluid Mechanics 2014 46:1, 23-45).**

**Author response:**

Thank you for drawing our attention to this publication.

**Change to manuscript:**

Following Page 5 lines 20-21 "All of the laminated haze layer reports are from aircraft campaigns of short duration, and all excluded from consideration any measurements which included ice crystals and clouds.", we insert a new paragraph:

**``In mid-latitude examples of extremely strong atmospheric boundary layer stability, striations of fog may be identified at scales smaller than 1-metre (Mahrt 2014, Fig. 3). These are qualitatively similar to the cloud laminations identified by CRL. Perhaps the two phenomena share similar properties, particularly in terms of the factors which enable the laminations/striations to persist.''**

Text referring to Mahrt 2014 is also added to the new Section 5.6 ``Suggested explanations for the laminated phenomena.'' in the response to Reviewer 2 Comment 1, above.

New reference:

[Mahrt2014StablyStratBoundaryLayers]
Mahrt, L.: Stably Stratified Atmospheric Boundary Layers, Annual Review of Fluid Mechanics, 46, 23–45, 2014

**Referee Comment 4: What are the wind conditions here? Wind profiles and Richardson numbers would be a useful addition, and potential temperature profiles would also be more instructive that the included temperature profiles.**

Author response:

The twice-daily Eureka radiosondes provide windspeed and direction, and we have calculated potential temperature from the sonde temperature profiles as well. These have been added to Figures 5, 7, and 12, with accompanying text.

Richardson numbers (Ri) have not been added to the manuscript. We have calculated Bulk Richardson numbers using radiosonde data, but they are not particularly useful given the scope of this particular paper because the applicability and interpretation of these numbers is nuanced. The issue of turbulence vs. stability could be important, but more specific measurements in this area (e.g. aircraft with a turbulence probe) would be a more appropriate way to study this in detail, in the future.

In general, interpretation of Richardson Numbers smaller than some critical value Rc is that the atmosphere is dynamically unstable and favourable for turbulence, while at values greater than the critical value, it is interpreted to be stably-stratified ("turbulence cannot be sustained", but is also not precluded entirely). The exact Ri values calculated depend on the vertical resolution of the profiles used to make them (Stull 1988, Balsley 2008, Tjernstrom 2009), and so does the value for Rc. Rc can vary from 0.25 (Stull 1988 p. 177) to Rc = 1 or more (Shupe 2017, who uses a minimum cutoff of Ri=1 to guarantee nonturbulence, while still allowing for exceptions of "weak, sporadic" turbulence). The larger Rc values are required for data which is lower resolution and/or smoothed. In our case, if we smooth, or if we choose an inappropriate Rc, we may miss some small patches of instability. This might not be tolerable considering that we are examining laminations at 7.5 m resolution.

As an example, in the figure to the right for 21 March 2017 11:00 UTC, the blue line gives the result at the maximum sonde resolution; the black line gives the result when the windspeed and temperature profiles have been smoothed first by a 3-point moving average filter. We have 238 unsmoothed instances of Ri<1 , and only 139 smoothed instances of Ri<1.

[Figure]

Further, there is a known hysteresis effect in laminar flows, whereby the Richardson Number may begin larger than the critical value (i.e. is stable), then drop below the critical value (there becoming turbulent), and then rise again above the critical value, yet not reaching stability again until a much higher value is reached (Stull 1988 in Brooks 2017). Gravity waves are another example in which turbulence can exist at high Ri. Therefore, interpretation of Ri values we may calculate is also nontrivial.

In order to properly address turbulence/stability, we should also consider whether the dry Ri indicated above are applicable to our situation within clouds. Brooks 2017 advocates the use of such dry Ri (Ri_d; calculated as in Brooks 2017 Eqn 1b, based on Stull 1988 Eqn 5.6.2) only in the case of cloud-free air. They indicate that moist Ri (Ri_m; calculated as detailed in Brooks 2017 Eqn 2, using equations based on Durran&Klemp 1982 Eqn 5) are more appropriate in liquid and mixed phase clouds. To use the latter equations, data contributed from a microwave radiometer or similar is required

- which is far outside the scope of what we can provide for the present manuscript.

Because Richardson Number is a quantity which requires such careful and nuanced calculation and interpretation, the authors did not feel that the current paper was the appropriate place to address into this topic. The Richardson numbers that we calculate at this stage only confound the interpretation of the laminations, while other profiles (wind, temperature, etc) are more straightforward and instructive. To fairly cover the topic of stability would require such space in the paper that it would detract from the main point of the manuscript: the demonstration of laminations within Arctic clouds.

The authors agree that a formal assessment of atmospheric stability in the context of these laminations is an important avenue to pursue in follow-on papers.

New references (just for review response; not required in paper):

[Brooks2017TurbulentSummerBoundaryLayer]
Brooks, I. M., Tjernström, M., Persson, P. O. G., Shupe, M. D., Atkinson, R. A., Canut, G., Birch, C. E., Mauritsen, T., Sedlar, J., and Brooks, B. J.: The turbulent structure of the Arctic summer boundary layer during the Arctic Summer Cloud-Ocean Study, Journal of Geophysical Research: Atmospheres, 112, 9685–9704, 2017.

[Stull1988]
Stull, R. B. (1988). Introduction to Boundary Layer Meteorology (pp. 666). Dordrecht, The Netherlands: Kluwer Academic Publishers.

[Stull2017PracticalMeteorologyBook]
Stull, R.: Practical Meteorology: An Algebra-based Survey of Atmospheric Science, Roland Stull, The University of British Columbia, Vancouver, Canada, 1.02b edn., 2017.

[DurranKlemp1982MoistureBruntVaisala]
Durran, D. R. and Klemp, J. B.: On the Effects of Moisture on the Brunt-Väisälä Frequency, Journal of the Atmospheric Sciences, 39, 2152–2158, 1982.

[Balsley2008GradientRichardsonNumber]
Balsley, B. B., Svensson, G., and Jjernström, M.: On the scale-dependence of the gradient Richardson number in the residual layer, Boundary- Layer Meteorology, 127, 57–72, 2008

[Tjernstrom2009VerticalArcticTropoERA40]
Tjernström, M. and Graversen, R. G.: The vertical structure of the lower Arctic troposphere analysed from observations and the ERA-40 reanalysis, Quarterly Journal of the Royal Meteorological Society, 135, 431–443, 2009.

**Change to manuscript:**

Figures 5, 7, 12 have been modified to include potential temperature, windspeed, and wind direction plots.

The text fromPage 9 line 30 through Page 10 line 8 (original version numbering), and Page 11 line 9 through  Page 11 line 27 (original version numbering), have been changed to address the modified figures.

---

## Author Comment (AC2) · 29 Dec 2018

**Major Comments**

**Referee Comment 1: Figure 2: There appears to be very little attenuation of your light within this cloud. This raises concern for me about multiple scattering enhancing your signal. The Nott et al. 2012 paper described the field of view of the system at 0.3-2mrad. What were you running for this data collection? Is multiple scattering a concern? 3d multiple scattering effects could be very difficult to separate from physical structure and could add (or smooth out) structure on the order of a few range bins depending on the physical features of the cloud.**

**Author response:**

The lidar is run in operational mode at 1.5 mrad field of view.

Test runs with fields of view of 0.5 and 1.0 mrad during the same type of meteorological conditions as those shown in the paper, with laminated clouds extending to about 5 km, have shown that the laminated features remain in the measurements.

The laminated clouds in general are not always particularly optically thick. Therefore, the range-scaled photocount returns are not always much lower at the top edges of the cloud compared to at the lower edges. Recall that all of our plots have been range-scaled. They do attenuate the light overall, as we can see in the 11 November 2017 example: From 1-9:00 UTC, the clear air above the cloud has range scaled count rates $< 10^6$ MHz m^2. From 9:00-24:00 UTC, the count rates at those altitudes is much higher, at $10^8$ MHz m^2.

We do not believe strong multiple scattering to be a major issue here, as the major point of the manuscript concerns the detection of the laminations in the clouds. It is unlikely that the laminations can be explained away by this mechanism.

Multiple scattering is always of concern for any lidar measurement which goes through optically thick clouds. Consider the example from 26 August 2017 (now moved to Appendix ``A summer example of layers on 26 August 2017''; Fig 11), before 4:30 UTC. The optically thick cloud at about 2 km stops nearly all signals from penetrating past that altitude. Multiple scattering would surely be something to concern ourselves with in the upper reaches of the parts of the cloud that we can examine there. Later in that same measurement, after 5:00 UTC, we note that the 2 km cloud has dissipated or moved away, leaving a cloud much thicker in vertical extent, but much thinner in terms of optical properties, for the next hour. Looking above *that* cloud, we again see that the laser beam eventually gets attenuated - but not until 4.5 km or so. We might consider multiple scattering to become important in the upper reaches of the cloud: Particularly later in the measurement, after 6 UTC. That said, it is unlikely that multiple scattering is of considerable concern between 5-6 UTC at the lower altitudes, and there are plenty of laminations present below, say, 3 km. It should be of less concern when the beam penetrates entirely through the cloud without being fully attenuated.

If multiple scattering were present, we would expect its effects to increase (a) with penetration depth into the cloud (because of more integrated material to be scattering off of), and (b) with altitude (because it is geometrically easier to multiply scatter photons in if they originate (originally scatter) farther from our lidar).

A helpful indication that multiple scattering is not the sole cause of these layers is the depolarization measurements. Returns which are multiply scattered would tend to have depolarization parameters (and thus depolarization ratios also) of approximately 1. We do not see any general trends with altitude, nor indeed any positive correlation with overall local count rate, tending toward higher depolarization. Therefore we find that multiple scattering is probably not a major concern for the detection of the laminations in the clouds we observe.

Multiple scattering is something that we can look into more fully in future. Some numerical studies to determine precisely what geometric effects we could expect, for example. Any influence which may yet exist from multiple scattering does not detract from the detection of the laminations in our measurements at their most basic level - it is certainly unlikely that multiple scattering would be accountable for all of the laminations at all altitudes including the lower ones.

**Referee Comment 2: You say several times that taking data at lower resolution would cause the thin features to be covered (example on Page 2, Line 19-20). I am skeptical that this would completely remove some of the features you see, though I do not doubt it will change them. For example, the thick count layer at 3km from 4-5.5 UTC in Figure 1 would possibly remain. I believe you should show high vs. degraded resolution to better illustrate this point. Further, it will allow you to quantitatively assess, both what other investigators should be looking for in their lower resolution data and define to what extent data is masked. Specifically, it would help place your work in the context of the previous authors you describe on Page 2, Lines 28-33. Additionally, it will suggest how fruitful further analysis might be, combining data with the low-resolution lidar data products.**

**Author response:**

We will add such a degraded resolution plot to illustrate the point.

**Change to manuscript:**

We have added a new **Figure 3** to address this. It is comparable to Fig 1 and Fig 2, but shows data at 1 min x 75 m resolution.

Text has been amended to: "If the data are averaged to altitude bins 10 times as large as those shown, all traces of the laminated structure would be erased **(Fig. 3), and the cloud would look more similar to** a smooth cloud."

**Referee Comment 3: Why do you not apply overlap corrections? Showing data below 500 meters and not overlap correcting is confusing to me and a bit misleading in places. Suggest either applying the corrections or removing all data below full overlap for clarity.**

**Author response:**

We have removed all lidar data below 500 m. Our overlap correction routines are still in development.

**Change to manuscript:**

Figures including lidar measurements have been modified to include only data > 500 m. New figures

for quantities unaffected by lidar overlap (e.g. sondes) include all altitudes for context.

**Referee Comment 4: Page 10, Lines 27-29 and Figure 3 and Figure 4: Depolarization contours are very noisy. I would argue they are almost unhelpful. In fact, given the results of the McCullough et al. (2018) paper, I am questioning if you have the sensitivity in the depolarization channel to make the described measurements at 1 minute resolution. At the very least, contours of depolarization error bounds should be shown to inform your reader how far they can trust the interpretation of depolarization.**

**Author response:**

We agree that the depolarization results given in this manuscript are noisy, and are not conclusive of much on their own. They are calculated using the d1 method, which uses our low-count rate perpendicular channel, which we would typically run at 20 minute x 37.5 m resolution. To do the d2 method (three-channel) is more difficult to calibrate, and was not done for these dates. Not least because we have realized that there are relevant morphological (and perhaps depolarization-sensitive) features at the highest resolution scales. So to use d1 at low resolution (getting d1 values equal to an average of the layers and inter-layer values, perhaps producing d1 values which do not actually exist anywhere in the cloud!) to calibrate d2... it brings a level of complexity that we could not sufficiently explore given the scope of the current document.

Considering the success we had with the range-scaled photocount profiles at the highest resolutions, we thought it worthwhile to include our d1 values (since we *know* what we're calculating in that instance, although it's noisy) at the same resolution and see what happens.

We might have expected smoother (more uniformly noisy?) d1 colour plots which would have indicated nothing at all. However, as in Fig 4e, we do see that we have enough information to determine (a) no strong correlation of high depolarization with strong laminations (and in fact, anti-correlation seems more likely), and (b) variations in depolarization which do seem to be correlated with fall streaks.

**Changes to manuscript:**

We have added a new appendix: ``Depolarization Uncertainty'', with the following text:

``For completeness, depolarization uncertainties for the two main dates examined in this paper are presented here. Figure 9 for 21 March 2017, and Fig. 10 for 11 November 2017.''

Include two new figures **Fig. 9 and Fig. 10.**

In the existing body of the paper:

At page 7 line 9, add the new text: ``**Examples of depolarization parameter plots are Figs. 4e and 6e. Appendix A provides some plots of depolarization uncertainty in Figs. 9 and 10.''**

At Page 10 line 31, add the new text: ``**Although the depolarization plots are somewhat noisy at this resolution, absolute uncertainties are generally between 0.05 and 0.1 (in the same units as depolarization parameter) for the region below 1 km, where the laminations are visible in Fig. 4e. At higher altitudes, uncertainties for this date reach 0.16.''**

At Page 14 line 32, add the new text: ``**For Fig. 6e, the uncertainties are somewhat higher than they are for 21 March 2017 (4e) in regions of high depolarization, reaching values of 0.2 to 0.3 where d>0.5. Similar to the March example, regions on 11 November 2017 in which cloud laminations are visible, namely between 10:30 and 11:00 UTC below 1 km, have absolute uncertainties smaller than 0.06 in general.**''

**Referee Comment 5: Section 4.3: Perhaps this is best used as an appendix. It is less convincing than the other 2 cases based on the level of information you are able to provide. It might be more helpful to summarize your measurements to describe the percent of time you see clouds with such vertical laminations.**

**Author response:**

We will move this summer example to an appendix. See response to comment 7b in this document about the percentage of time we see clouds with the laminations.

**Change to manuscript:**

Content from old Section 4.3 has been moved to Appendix C: ``A summer example of layers on 26 August 2017''.

**Referee Comment 6: Section 5.1-5.3: The discussion in these 3 sections is a major weakness of the paper in my opinion. I do not find the discussion particularly convincing because the topics discussed, while likely being familiar to a reader knowledgeable of lidar hardware, is not particularly well constructed in my opinion.**

**Author response:**

The general construction of this section was dictated by questions that we have received when showing the laminated cloud measurements to colleagues and at conferences. Analagous to R1's Comment 1 ("Could this be a multiple scattering effect?"), the questions addressed in Sections 5.1 - 5.3 show the concerns of those people whose immediate impression is that these laminations might be a result of instrument or measurement effects or artifacts. The authors interpret the laminations to be geophysical, but this is because we have good reasons for believing them not to be instrumental effects, as detailed in this section. Detailed responses follow, but we can be more explicit in the manuscript in explaining why this section exists in the format it does.

**Change to manuscript:**

Following the sentence "Before attributing the striped effect that we see in our data to geophysical phenomena, we apply due diligence to show that it is not an instrumental effect.", we add the new sentence "**Each of the topics covered by Section 5.1 - 5.4 address a specific instrumental or measurement effect/artifact which has been suggested by members of the broader lidar community as a possible indication that the laminations are not geophysical phenomena.**"

**R1 comment 6, continued: My concerns are as follows:**

**6a) PMT or saturation more generally should serve to smooth your profiles in every case I can imagine. If the section of your glued profile originates from photon counting data, photons will be under-reported and thus thick clouds will seem thinner. If the portion of the profile is from the analog counting system and you are under reporting intensity (or even clipping the ADC), you are operating so far outside of the designed regime of the detectors that the data is likely not valid. Additionally, you claim to have corrected it in Section 3.**

**Author response:**

We have no saturated measurements in the paper. Indeed, as R1 points out, it would not be appropriate to include saturated measurements in our analyses in the first place. One sentence explaining that we are not operating near saturation limits for our system should suffice to stave off this line of questioning. Regardless, these logical arguments seem to be not quite as convincing as including a plot which shows the laminated features remaining, even at a factor of 10 lower count rates. Figure 6 (now Figure 8) was an easy test to carry out, and the results are visually convincing. We have changed the beginning of this section to more clearly make the point in words within the Discussion section.

**Change to manuscript:**

The first paragraphs of this section now read:

**"As discussed briefly in Section 3, the analyses are made using glued count rate profiles, which make use of photon counting signals in regions where the photon count rates are linear, and equivalent analogue signals in any region for which the photon counting rates become nonlinear. During routine processing, regions in which the analogue signals meet or exceed the counting limits of the analogue-to-digital converter are excluded from the retrieved profiles. For all measurements in this manuscript, the PMTs were not being operated near their maximum analogue count rates, so the likelihood of the laminations being PMT saturation artifacts is low.**

**Further, any saturation effects should serve to smooth out the profiles at high count rates, rather than inducing the oscillating count rates as we observe as the laminated cloud phenomena. In order to clearly demonstrate that these laminated features persist at much lower photon count rates, we performed a measurement with the aid of neutral density filters to lower the signal levels."**

**6b) Signal induced noise should be slow (microsecond time scale) and extensive in altitude.**

**Author response:**

Agreed.

**Change to manuscript:**

We have removed the mention of signal-induced-noise in the first paragraph of section 5.1.

**6c) PMT ringing on the other hand is something I would think could cause vertical structure on the scale described. I would think this is the major instrument effect to investigate.**

**Author response:**

We agree that PMT ringing could, under the right circumstances which we do not believe to be the case here, produce repetitive vertical structure in lidar data on the scales described. However, (a) we would not expect to see PMT ringing if the PMT is not being saturated (covered in 6a, above - our PMT is not saturated), and (b) we would expect the effects to be different than what we see in the cloud measurements: In the case of classical PMT ringing, we expect a signal which starts at very high count rates, repeating higher-than-surrounding-values at regular altitude intervals, and amplitude damping out with height. In our case, the laminations look quite different to that description. Even in the event of some PMT ringing (which we do not believe to be present at all here), during which some residue of the ringing signature is combined with the geophysical results above, but it would be insufficient to explain all of the laminated features we see in our cloud measurements.

The new Section 5.2 has some specific explanations, including a comparison to a figure from Kovalev and Eichinger 2004.

**Change to manuscript:**

We have added a new subsection 5.2 to specifically address PMT ringing, entitled "Ruling out PMT ringing".

New reference:
[Kovalev and Eichinger 2004]
Kovalev, V. A. and Eichinger, W. E.: Elastic Lidar: Theory, Practice, and Analysis Methods, John Wiley Sons, Inc., Hoboken, New Jersey, 1 edn., http://gen.lib.rus.ec/book/index.php?md5=16F1687DEAF30CDD0E02BC46D0453F58, 2004. pp. 122 - 123, Figure 4.6

**6d) I agree with your conclusion about laser power fluctuations. So much so that I would likely not even mention it in this analysis.**

**Author response:**

We agree that it seems almost too obvious to mention, however this question has come up in every presentation of these plots to the atmospheric community. Upon short reflection, all of those asking the question could see that "of course!" laser power fluctuations cannot be causing the laminations, but it is one of the questions which has been ubiquitous in discussions. Further, an explicit statement that laser power fluctuations are not an issue for qualitatively detecting the laminations provides support for range-scaled photocount profiles being sufficient for the purposes of this manuscript (i.e. calibrated backscatter coefficient profiles are not requisite for the detection of the laminations). Therefore, the authors would prefer to mention it in the text if the referees can accept it remaining there.

**6e) I agree mostly with your timing electronics conclusions but if you have an issue, it might not be stable in altitude. If you have 2 or more different clock speeds (from triggering, seeding, q switching, or your counting system clock drifting slightly), you could possibly alias one rate onto the other making your observations move in altitude. That would likely be a systematic shift observable at all altitudes though, and as such easy to identify.**

**Author response:**

This is an interesting point which we had not previously considered. We'll keep it in mind for future analyses. The effects that we see do not seem to be systematic shift at all altitudes, so it is probably not the case here.

**Referee Comment 7: I am surprised that the authors have not included lidar data that could be very helpful. They do call for more analysis in the conclusion. That said, without this analysis, I am not convinced that this work is a major observational contribution. Some omissions that I believe should be seriously considered (at minimum) are:**

**7a) I find myself surprised that the authors use radiosondes and not rotational Raman measured temperatures and vibrational Raman measured water vapor. This is especially true of Figure 4 where the thermodynamic structure changes dramatically over the observation period. The data need not be at 1 min resolution to be helpful.**

**Author response:**

We would have loved to use both the Rotational Raman temperatures and Water Vapour measurements from CRL for this analysis for precisely the reasons pointed out by R1, however it was unfortunately not possible for this study. Major funding cuts to CRL's research program several years ago have prevented us from addressing the issues which came up with both Rotational Raman temperatures and Water Vapour:

Since the initial testing results of the Rotational Raman Temperature channels indicated in Nott 2012, we have found that the laboratory temperature cannot be sufficiently tightly controlled to produce reliable temperature measurements. The interference filters for the relevant channels must be controlled to within +/- 2 degrees C in order for the results to be meaningful, and this is something we cannot accomplish with our current setup. Thus we're unfortunately limited to non-lidar temperature profile results, and hence use radiosondes.

Similarly, our water vapour channel has not been continuously operational for the duration of the laminated cloud measurements. Additionally, the water vapour results from Rotational Raman techniques, as applied to CRL, are only fully valid in clear skies. As we are looking at clouds, and sometimes optically thick clouds, these results would not help as much as we might wish.

We will be interested to use results from other Eureka water vapour measurement instruments in the near future.

**7b) I also find myself surprised that basic summary statistics of occurrence frequency or bounding relative humidity or temperature are not provided. At minimum, I would expect to see some observational bounds on conditions described in Section 5.4.**

**Author response:**

Determining the statistics is outside the scope of this phenomenological study.

We intend to continue this project by exploring the frequency and distribution of such laminated clouds throughout the year. An intermediate step is to determine objective criteria by which we can determine whether a given time period of CRL data exhibits the required characteristics to be included vs. excluded from the population of laminated clouds. (How thin do the laminations have to be to qualify?

How many layers are required in a vertical sample? What amplitude in signal must these laminations have?). Likewise, we must determine a course to account for dates with no lidar measurements, and dates for which the lidar beam is attenuated at low altitudes - both being cases which do not preclude the existence of laminated clouds, but which would not be counted as a detection of them, either. These are not trivial tasks, so including a hard percent value for what percent of the time we see these clouds would be, at this stage, premature. Therefore in the current manuscript, we aim instead to simply point out that this laminated cloud phenomenon is not limited to wintertime measurements at Eureka.

We will make a comment regarding frequency at the start of Section 4: "Results".

Change to manuscript:

At the start of Section 4, insert the following text:

``**CRL made 182 days of measurements between March and December 2017. Of these, at least 45 days show evidence of horizontal laminations within clouds. Thus, laminations occurred on 25 % of all measurement dates. A minimum of one detection of laminations was present in each measured month. Hence, this phenomenon is not restricted to a particular season. March 2017 had highest rate of detections, with at least 10 of 24 measurement days demonstrating laminations. Three representative examples will be shown in full here: 21 March 2017 is in Section 4.1, 14 November 2017 is in Section 4.2, and 26 August 2017 is in Appendix C.**"

**7c) I am not sure raw photon counts are sufficient to quantitatively show the structures within clouds. Calibrated backscatter coefficients would be much more useful. Additionally, they remove uncertainty sources such as laser power fluctuations.**

**Author response:**

Calibrated backscatter coefficient measurements require a normalization in clear air (or air of known aerosol backscatter cross-section for each measurement period. Typically, the region for this clear air is taken above any clouds and aerosol layers which are present. However, the clouds studied in this manuscript often nearly obscure any photons from heights above the clouds. At a minimum, the top parts of the clouds are likely to exhibit multiple scattering, and thus we cannot be sure of the returns above these levels. Likewise, a normalization region below the clouds is typically not available for these cloud examples, most of which extend down into our overlap region. Therefore the normalization for these dates is difficult.

Further, with CRL's SNR, we are unable to calculate calibrated backscatter coefficients at sufficiently high resolution to resolve these layers - this is presumably one reason that we had not noticed the laminations previously. The operational resolution for routinely retrieved CRL calibrated backscatter coefficients is 10 minutes x 105 metres.

Now that we have some motivation to examine the CRL data at high altitude resolution, we are investing further efforts into producing the best backscatter coefficient profiles we can.

We agree that raw photon counts are not ideal for a quantitative analysis of the amplitude of these laminations. However, given that no published works have, to our knowledge, done so much as to point out the existence of these laminations, we felt that publishing our findings that these laminations *exist at all* was important.

As pointed out by R1 in comment 7b, there are other quantitative results we can provide going forward, even with the raw counts profiles: Statistics about the occurrence rates for these features, and similar.

As also pointed out by R1 in comment 7d, laser fluctuations are not capable of producing false horizontal laminations in the plots. Given that this is the first paper to demonstrate the existence of these laminations in lidar data such as CRL's, we prefer to get the finding out into the community for further discussion as soon as possible (i.e. using range-scaled photocounts), and follow this up with calibrated backscatter profiles as becomes possible. There is lots of interesting quantitative follow-on work which should be pursued - and for that, we will surely address the effect of laser power from the measurements as much as we can.

Finally, the range-scaled photocounts presented here have been saturation/deadtime corrected, background corrected, PC and Analogue signals have been glued into a merged profile, and the plots are thus not quite raw profiles in any case.

**Minor Comments:**

**Referee Minor Comment 1: Page 1, Line 4: It obviously depends on your target but 1 min time resolution might not be particularly high resolution. Suggest dropping the word "high" here. Also on Page 5, Line 10**

**Author response:**

Done.

Our target is stable over a several minute period, so our measurements have high enough time resolution to detect these. We did want to make the point that observations at 20 minute time resolution, for example, are not as helpful - but it's true that lidars such as that in Hayman et al 2012 have much higher time resolution by a factor of over 100x.

**Change to manuscript:**

Removed the word "high" to make the sentence: **"CRL's time (1 min) and altitude (7.5 m) resolution ... "**

**Referee Minor Comment 2: Figure 2: At 3km, the range correction should be 9X 10^6. The counts that you are showing are therefore on the order of 10-100. Is that correct? If so, counting statistics worry me. It is impossible to tell here what wiggles are due to scattering phenomena and what wiggles are due to counting statistics. Suggest adding error bars to clarify.**

**Author response:**

Yes, that range correction is correct, but the count rates are higher. The data are shown in MHz. The number of counts per measurement bin per unit time in this plot ranges from just under 600 photons/altitude bin/minute near 4 km, to over 10000 photons/altitude bin/minute at 3.15 km. The photon counting mode is used when raw signals are smaller than an equivalent to 20 MHz (N = 600 photons/bin/min) and the analogue mode is used above that. The uncertainties for the analogue mode include both poisson noise (approx. sqrt(N)) and systematic uncertainty introduced by the ADC

converter.

Here is the same data, shown in units of photocounts/bin/minute (not MHz), with shaded error bars, without altitude scaling, shown on a linear x-axis. The blue profile is located at its true position on the x-axis. The red, black, and green profiles are offset to the right of their true values by 5000, 10000, and 15000 photocounts respectively. The uncertainties are far smaller than the magnitude of the wiggles:

[Figure]

The wiggles due to counting statistics would be a larger worry if we had only one profile, and/or for channels which have fewer photons (e.g. depolarization perpendicular channel). However, statistical counting errors are likely to appear as white noise - as likely to be above as below the "true" profile. They should not be correlated in time for any given altitude. Given that we have profiles showing wiggles which *are* correlated in time, we consider that these are more likely to be due to scattering phenomena, and not counting statistics.

**Change to manuscript:**

We have added error bars to Figure 2 to indicate the extent of the uncertainty.

We have added the bolded text to the Figure 2 caption:

"Selected profiles of range-scaled 532 nm photocounts as a function of altitude for four consecutive

minutes just after 06:40 UTC on 7 March 2016 (same date as for Fig. 1), each offset by $1 \times 10^{0.6}$ (or 4 m$^2$MHz) along the x-axis, between the altitudes of 3 to 4 km. **Shaded areas show uncertainty.** There are clearly horizontal coherent structures in the cloud in space (aliased to time by motion over the lidar) at least down to the 7.5 m height resolution of the lidar."

**Referee Minor Comment 3: Page 3, Line 1-8: The following paper and references therein may be of interest to the authors as motivation for cloud structure size scales: Beals, et al., "Holographic measurements of inhomogeneous cloud mixing at the centimeter scale," Science 350, 87–90 (2015).**

**Author response:**

Thank you for drawing our attention to this paper.

**Changes to manuscript:**

We have added a paragraph about this reference starting on Page 4 Line 3 (new version of manuscript). Following the paragraph which reads: "We have been unable to find many references to cloud features at sub-100m scales in the literature... Again, these situations are quite different in morphology from the laminated features described in this paper.", we add the new text:

**"Measurements by airborne holographic imaging have visualized the spatial structure in clouds at centimetre scales by measuring droplet size and number distributions, revealing that clouds are inhomogeneous and contain sharp transitions between cloud and clear air properties even at the smallest turbulent scales (Beals et al. (2015)). Given that there are ``edges'' within clouds even at cm scales, it is reasonable to infer that there may be structural cloud features which are relevant to the overall interpretation of particular clouds, which are possible to investigate by lidar at resolutions of tens of metres and which will be missed entirely by lidar measurements at 100+ m scales. Certainly, the scales probed in Beals et al. (2015) are significantly smaller than those possible to investigate using the CRL lidar. Cloud measurements covering the entire range of spatial scales from centimetre to global is ultimately required. CRL helps close the gap from over four orders of magnitude of spatial size, to three, between the holographic imaging measurements and the smallest features currently discussed in the lidar literature."**

We have also made reference to Beals2015 in the new Section 5.6 ``Suggested explanations for the laminated phenomena''. See response to Reviewer 2, Comment #1, for the new text in that section.

New reference:
[Beals2015CloudHolography]
Beals, M. J., Fugal, J. P., Shaw, R. A., Lu, J., Spuler, S. M., and Stith, J. L.: Holographic measurements of inhomogeneous cloud mixing at the centimeter scale, Science, 350, 87 – 90, 2015.

**Referee Minor Comment 4: Page 4, Line 17: Referring to a broad class of elastic scatter lidars as Mie lidars is very imprecise. Suggest modifying to "elastic scatter" as you have no way of verifying that all scatterers are spheres.**

**Author response:**

The lidar which was deployed to Alert was explicitly called a "Mie Lidar" in Hoff 1998. The term "Mie

Lidar" in our sentence on Line 17 (old version) refers only to that particular lidar.

**Referee Minor Comment 5:  Page 6, Line 26: This sentence is confusing because your lidar counting system has already binned single photon data to 7.5 meters and 1 minute. Suggest modifying this sentence to something like: "No further binning was performed. "**

Author response:

We have made this change.

Change to manuscript:

The sentence has been modified, as suggested by R1, to: "**No further binning was performed**".

**Referee Minor Comment 6:  Figures 3 and 4: I believe there are several ways to calculate relative humidity with respect to ice. There are several parameterized versions or more simple versions. They do not all result in identical values given identical inputs. Suggest adding a citation to describe the method you use.**

Author response:

The Goff-Gratch formulation has been used for calculations.

Change to manuscript:

Appendix B has been added to the manuscript, which reads:

**Appendix B: Calculations of RH over ice**

**Relative humidity with respect to liquid water (RHw) is converted to relative humidity with respect to ice (RHi) using the Goff-Gratch formulations for saturation vapour pressure (Goff and Gratch (1946), in List (1949)). Saturation vapour pressure over water, ew, can be calculated via equation B1:**

$$\log_{10} e_w = -7.90298 \left(\frac{T_s}{T} - 1\right) + 5.02808 \log_{10}\left(\frac{T_s}{T}\right) - (1.3816 \times 10^{-7})(10^{11.344\left(1-\frac{T}{T_s}\right)} - 1)$$
$$+ (8.1328 \times 10^{-3})(10^{-3.49149\left(\frac{T_s}{T}-1\right)} - 1) + \log_{10} e_{ws},$$

**in which T is the radiosonde temperature in Kelvin, Ts = 373.16 K is the steam point temperature of liquid water, and ews = 1013.246 mb is the saturation pressure of liquid water at the steam point temperature (at 1 standard atmosphere). Saturation vapour pressure over ice, ei, can be calculated via equation B2:**

$$\log_{10} e_i = -9.09718 \left(\frac{T_o}{T} - 1\right) - 3.56654 \log_{10}\left(\frac{T_o}{T}\right) + 0.876793 \left(1 - \frac{T}{T_o}\right) + \log_{10}(e_{io}),$$

**in which To = 273.16 K is the ice point temperature, and eio = 6.1071 mb is the saturation pressure of ice at the ice-point temperature (at 0.0060273 standard atmospheres). Relative humidity with respect to ice, in percent, is then equation B3:**

$$RH_i = \left(\frac{e_w}{e_{io}}\right) RH_w$$

New references:

[GoffGratch1946LowPressureWater]
Goff, J. A. and Gratch, S.: Low-pressure properties of water from -160 to 212 F, in: Transactions of the American society of heating and ventilating engineers, pp 95-122, 52nd annual meeting of the American society of heating and ventilating engineers, New York, 1946.

[ListSmithsMetTables1949a6thEd]
List, R. J.: Smithsonian Meteorological Tables, vol. 114 of Smithsonian Miscellaneous Collections, Smithsonian Institution Press, 4th reprint (1968) of 6th revised edn., 1949.

**Referee Minor Comment 7: Figure 4 Caption: Suggest shortening by describing panels a-f as "same as Figure 3" or similar.**

**Author response:**

Done.

**Change to manuscript:**

We have revised the old Fig 4. caption (now numbered Fig. 6) text to read: **"Measurements from 14 November 2017; (a-f) same format as Fig. 4. Thick clouds were present early in the day, with cloud cover reducing later. Layers which start in a cloud continue in the next section of cloud, even if there is a gap in between. Precipitation alternated between light snow, blowing snow, ice crystals, and no precipitation at the ground throughout the day."**

**Referee Minor Comment 8: Page 9, Lines 27-29: Low depolarization is consistent with observations of preferentially oriented ice crystals. Suggest clarifying that high depolarization is ": : :inconsistent with interpretation as randomly oriented ice particles." Note that the following might be of interest as well, especially Appendix A: Silber, et al., "Polar liquid cloud base detection algorithms for high spectral resolution or micropulse lidar data," J. Geophys. Res.: Atmos. (2018).**

**Author response:**

We will clarify that low depolarization is inconsistent with randomly oriented ice particles, but is consistent with preferentially oriented ice particles.

**Change to manuscript:**

We have added text, as bolded here:
"The 45 m thick layers are displayed with a high depolarization parameter, which indicates non-spherical particles. Typically, this means **randomly oriented** frozen particles within clouds, or aerosol particles outside of clouds."

and

" The depolarization values in these regions are low and therefore combined with the high backscatter signal are consistent with liquid water droplets **and/or preferentially oriented ice particles**, and are inconsistent with interpretation as **randomly oriented** ice particles."

**Referee Minor Comment 9: Page 10, Line 15 and elsewhere: I find the use of numbers like 1X10^10.5 to be difficult to interpret. Suggest changing to integer powers: 1X10^10.5=3X10^10 or much less preferably changing to dB.**

**Author response:**

The non-integer powers are included for direct comparison to the log colour scale in the plots. We have now added in the brackets a conversion to integer powers after each instance in the text.

**Change to manuscript:**

For Fig. 2, the caption and corresponding text now reads: ... each offset by $1x10^{0.6}$ **(or 4 m^2MHz)** ...

On Page 14, the brackets now read: ($1\times10^{10.5}$ m2MHz rather than $1\times10^{10}$ m^2MHz**; equivalent to 3.2x10^10 m^2MHz vs. 1X10^10 m^2MHz**).

On Page 14, describing new Fig 6 (old fig 4): ($1\times10^{8.8}$ **(or 6.2 x 10^8)**, red in Fig. 6b, and $1\times10^{8.5}$ **(or 3.2 x 10^8)** , yellow in Fig. 6c, respectively).

**Referee Minor Comment 10: Page 11, Line 16-17 and throughout the manuscript: I assume your sondes are reporting their raw data with respect to water. Are you reporting all relative humidity values with respect to ice? It is clear in the figures but less so in the text. Suggest adding "w.r.t ice" or "w.r.t. water" throughout the text to clarify or inserting a blanket statement specifying how all data are reported.**

**Author response:**

The sondes provide their raw data with respect to water. We then calculate the corresponding values with respect to ice where relevant (see response to R1 minor comment 6, above). We will clarify which RH is meant in each case in the text.

**Change to manuscript:**

Each instance of relative humidity in the text is now specified as with respect to water or with respect to ice in the manuscript.

**Technical Corrections:**

**Referee Technical Comment 1:  Page 10, Line 5: ": : :the air is remains: : :"**

**Author response:**

Done.

**Change to manuscript:**

Correction made to: "... the air remains..."

**Referee Technical Comment 2: Page 10, Line 14 and elsewhere: "The clouds[,] which contain: : :" The use of the word "which" requires use of a comma in most places.**

**Author response:**

Here we intended no comma. We want to say that the [particular clouds which contain the layers] are found below 4 km, to clarify that not *all* CRL clouds are found below 4 km.

We have corrected this issue where it comes up in other locations.

**Change to manuscript:**

None at this location; commas added where needed elsewhere.

**Referee Technical Comment 3: Page 19, Line 18: I believe the paper you refer to here is in the January 2012 publication, not 2011.**

**Author response:**

We have made the correction. That article was published online 10 Dec 2011 and we had afterward neglected to update the reference to the final January 2012 publication date.

**Change to manuscript:**

Reference now reads: **Morrison, H., de Boer, G., Feingold, G., Harrington, J., Shupe, M. D., and Sulia, K.: Resilience of persistent Arctic mixed-phase clouds, Nature Geoscience, 5, 11–17, 2012.**

---

## Author Response (AR2)

Thank you to the anonymous Referee #3 for their comments on this manuscript. We have addressed the Minor Comments as follows:

**Referee comment 1a: The justification supporting the purpose of the intended study is not well written.**

**Author response:** We have added some text giving the motivation for the study. It was an exploratory study, to find out whether there were any cloud features present at resolutions below the processing resolution of our usual data products. The results showed physical cloud phenomena (laminations) which had not been previously discussed in the literature.

**Change to manuscript:** New text added after Page 2, Line 5:

"Here, we have carried out an exploratory study of our lidar data at the raw measurement resolution (1 min × 7.5 m) to determine whether there are any cloud features in the data which should be accounted for when interpreting our lower resolution derived data products (typically 20 min × 37.5 m, to optimize signal-to-noise ratios). The results revealed laminated features in the Arctic clouds which we have not found to be discussed previously in the literature."

**Referee comment 1b: Also, the introductory aspect of the manuscript is weak in review of previous studies and lacks basic communication for any reader.**

**Author response:** We have separated the review of previous studies into its own section (the new Section 2: Literature Search). We have made the introductory text more concise, in hopes that this will ensure clearer communication.

**Change to manuscript:** New literature search section is now separate from introduction, which has been clarified. New text added to end of literature search section:

"The summary of the literature shows that hydrometeors seem to persist in laminated formations, but have thus far been shown only at coarser scales than that of CRL's measurements. Likewise, haze shows finer laminations. Many lidar systems either have sufficiently low measurement resolution, or data which is smoothed during processing to low enough resolutions, that the cloud laminations shown by CRL are absent from the resulting lidar measurements. "

**Referee comment 2: Rephrase Line 20 -25, the sentence is too long**

**Author response:** We have shortened the sentence, and made it clearer.

[revised manuscript text omitted]